# Leading Edge Technologies and Perspectives in Industrial Oilseed Extraction [note 1]

**DOI:** 10.3390/molecules28165973

**Published:** 2023-08-09

**Authors:** Christian Cravotto, Ombéline Claux, Mickaël Bartier, Anne-Sylvie Fabiano-Tixier, Silvia Tabasso

**Affiliations:** 1GREEN Extraction Team, INRAE, UMR 408, Avignon Université, F-84000 Avignon, France; christian.cravotto@alumni.univ-avignon.fr; 2Pennakem Europa (EcoXtract®), 224 Avenue de la Dordogne, F-59944 Dunkerque, France; oclaux.papers@gmail.com (O.C.); mickael.bartier@minakem.com (M.B.); 3Department of Drug Science and Technology, University of Turin, Via P. Giuria 9, 10125 Turin, Italy; silvia.tabasso@unito.it

**Keywords:** oilseed, enabling technologies, oil extraction, alternative solvents, protein meal, secondary metabolites, intensification

## Abstract

With the increase in the world’s population and per capita wealth, oil producers must not only increase edible oil production but also meet the demand for a higher quality and variety of products. Recently, the focus has shifted from single processing steps to the entire vegetable oil production process, with an emphasis on introducing innovative technologies to improve quality and production efficiency. In this review, conventional methods of oilseed storage, processing and extraction are presented, as well as innovative processing and extraction techniques. Furthermore, the parameters most affecting the products’ yields and quality at the industrial level are critically described. The extensive use of hexane for the extraction of most vegetable oils is undoubtedly the main concern of the whole production process in terms of health, safety and environmental issues. Therefore, special attention is paid to environmentally friendly solvents such as ethanol, supercritical CO_2_, 2-methyloxolane, water enzymatic extraction, etc. The state of the art in the use of green solvents is described and an objective assessment of their potential for more sustainable industrial processes is proposed.

## 1. Introduction

Vegetable oils represent a valuable class of bioresources used in both food and non-food industries, and their production has increased steadily over the past two decades. The demand for edible oil is increasing as the world’s population grows and purchasing power improves. The global production of vegetable oils has risen from about 90.5 million tonnes (Mt) in 2000/2001 to 207.5 Mt in 2019/2020, and this trend is expected to continue in the future [1]. Oilseeds account for 20% of global grain production, with soybean, rapeseed and sunflower oil being the most produced both globally and in France, with a total production of about 100 Mt in 2018/2019 (Table 1) [2]. Most seed oils are edible and used as end products for food applications. As the world’s population and per capita wealth increase, oil producers must not only increase their production of edible oil but also respond to the demand for a higher quality and variety of products. Moreover, the rapid increase in demand for biodiesel over the last two decades has put additional pressure on the demand for crude oil. The increase in global demand for edible oil has mainly led to an enormous expansion of cultivated areas for soy in South America and for oil palms in Malaysia and Indonesia. In addition, the selective choice of seeds in agriculture has improved the yield per hectare or tolerance to climatic fluctuations as well as the properties of the end products (oil and meal) [3]. Other oilseeds of lesser economic importance are often used as raw materials to produce food (e.g., chocolate, margarine) and non-food products (e.g., soap, biodiesel). As agricultural products, oilseeds undergo several mechanical, thermal, chemical and/or biochemical transformations, which are referred to as food processing. The purpose of food processing is to transform a usually inedible raw material into a food product in order to ensure the safety of the product and improve its preservation and shelf life [2].

Oilseed production consists, in principle, of four basic steps: seed storage, cleaning and preparation; oil extraction; processing of the pressed oil and/or miscella (oil-rich solvent); and processing of the cake or meal. Of the many steps involved in oilseed processing, oil extraction remains one of the most critical, as it determines the quality and quantity of the oil obtained. In recent years, however, the focus has shifted from individual operations to the entire production process. From this point of view, all operations are seen as interdependent, as the individual functions involved must now consider the impact of their actions on the overall process [5].

## 2. Oilseed Storage and Cleaning

The production of top-quality oils and meals requires the use of high-quality raw materials that are not damaged during transport, storage or processing. Seed storage is the step before the crushing and refining stages. Typical operations associated with storage include receiving, sampling, pre-cleaning, drying, storage and cleaning before processing [5], as shown in Figure 1. The main objective of storage is to keep the grain safe and deliver it to the processing plant at the right time and in the right condition.

Seeds arrive at the plant on an intermittent basis, by ship, train or truckload, in batches ranging from a few bags to loads of up to 60,000 tonnes. Each load of seed is controlled for weight and quality on receipt. Imported seed is usually also inspected on departure, while for locally harvested seed, the inspection at the plant may be the first it undergoes.

First, the received seed is weighed with weighbridges according to well-defined and fixed procedures to prevent fraud and ensure the absence of dead weights on entry and exit. The characteristics of the product, and consequently its economic value, can vary greatly depending on moisture, cracks, heat damage and other factors. When the product is received in storage plants, the quality of the raw material must be carefully and unequivocally controlled because it determines the type of processing and thus the costs, the type and the duration of storage. The moisture, protein content, oil content, presence of foreign matter and integrity of the seeds are periodically controlled, as deteriorated raw materials can cause serious capacity losses during refining. Crude oil obtained from damaged seeds (e.g., during harvest, transport or storage) may have a higher free fatty acid content, requiring further steps in the refining process, and exhibit increased oxidation resulting in a reduced stability [6].

The foreign matter (up to 2%) consists of naturally occurring sticks and husks, metals and stones accumulated during processing, as well as weed seeds and other grains, and is generally removed twice, once before storage and then at the beginning of the continuous process. The residual metals must be removed first to avoid any damage to the equipment and are magnetically separated using plate magnets, drum magnets and magnets suspended from a conveyor belt. Sticks and pods are generally larger and lighter than seed materials. Therefore, they are removed by both course screening and aspiration. Fine screening removes weed seeds, sand and soil. Stones are heavier than ground nuts and must be removed by gravity separation in machines called destoners [5].

The presence of impurities has a very negative impact on the sensory properties and chemical quality of oils, especially those obtained by mechanical pressing and not refined. Dimić et al. [7] showed that the aroma (smell and taste), the colour and the oxidative stability of mechanically extracted oils are negatively affected by the presence of impurities. Usually the impurities differ so much from the seed in terms of density and flow resistance that their separation does not pose any difficulties [8]. The contaminants removed by the magnet and scalper are normally disposed of as waste.

There are several factors to consider during storage, such as the humidity, oxygen, temperature and storage time. The temperature of the grain is monitored daily in order to immediately process the grain if heating occurs (temperature exceeds 40 °C). The rise in temperature is in fact seen as a precursor to qualitative damage to the seed due to the seed’s respiration processes triggering exothermic reactions. With a temperature of 70 °C, the seed is damaged within a few hours [5].

Storage stability can be affected by microbial, physical and chemical changes in the grain. For example, microbial growth can prompt spoilage, characterised by the development of sour tastes and unpleasant aromas, as well as deterioration of the physical structure. In addition, the growth of pathogenic microorganisms makes the products obtained from the grain unsafe. Water activity (a_w_) has become one of the most important intrinsic properties in predicting the survival of microorganisms in plant matrices and foods due to its direct influence on product quality and stability. In general, at a_w_ values below 0.61, bacterial proliferation is inhibited [9]. In addition, many types of chemical reactions related to grain deterioration have been studied as a function of a_w_ (e.g., hydrolysis, oxidation, enzymatic activity), showing that a_w_ is a key factor in predicting grain storage stability. For safe long-term storage, the moisture content must be reduced from 20% in the freshly harvested grain to less than 13% [5]. Moisture is localised in the non-fatty portion of the seed, where the critical moisture content reaches about 15%. The moisture content of the whole seed therefore depends on its oil content, e.g., the critical moisture content accounts for 9% for a seed with 40% oil and 8% for a seed with 47% oil [3]. Soybeans can be safely stored at a moisture content of 11% or less for 1 year or longer, while rapeseed and sunflower seeds require 8% or less. In general, a high moisture content in the seed (above 14–15% moisture) and high a_w_ accelerate the lipolytic enzyme reactions and increase the content of free fatty acids in the oils [10].

Grain drying can be performed using a vertical column system by direct fire, but today, air dryers are an essential part of modern storage facilities to preserve the quality of oilseeds [10]. Steam and even solar energy are also used as heat sources in some systems. The higher the drying temperature, the faster the drying process, but excessively high temperatures can have negative effects on the grain. For example, temperatures above 63 °C lead to a darker colour of the flour and oil, denature the proteins and increase the content of non-hydratable phosphatides in the crude oil [5]. Oilseeds are generally stored in vertical concrete or steel silos, which are usually galvanised, or in horizontal storage silos. Facilities that can hold seeds with high moisture content and store them for longer than 2 weeks should equip at least some silos with temperature-measuring devices and ventilation systems. However, ventilation can be more damaging than beneficial if it is not used properly. It is therefore important that fans are only operated when the air temperature is at least 3 °C cooler than the matrix and when the relative humidity is low [3].

## 3. Oilseed Preparation

Seed preparation involves a series of steps that make the seed suitable for extraction. Good seed processing results in high extraction yields (low residual oil content after extraction) and high-quality products (crude oil and meal) while keeping production costs low. In fact, the efficiency of oil extraction is strongly influenced by seed processing.

The cell structure of oilseeds contains hundreds of very small oil bodies (e.g., 0.5–3.0 μm diameter in rapeseed) anchored to the inner surface of the cell wall and the outer surface of the protein bodies [11]. Although oilseeds generally have a high oil content (20–50% *w*/*w*), the oil is firmly bound in the cell; thus, mechanical action is required to make it accessible for subsequent solvent extraction. However, the oilseed flakes, pellets or cake must be strong enough to withstand the impact of the solvent in percolation extractors but permeable enough for the solvent to penetrate their structure. The process presented here is a traditional method for oilseed preparation. Of course, each grain has its own characteristics, so different processes have been developed over the years. The main aims of the oilseed preparation include: weakening or breaking the walls of the oil-containing cells, increasing the oil extraction by pressing seeds with a high oil content before solvent extraction and shaping the material to facilitate solvent access to the oil [3]. Typical steps in oilseed processing include scaling, cleaning, cracking, conditioning (or cooking) and flaking. Depending on the process and the oilseed to be processed, drying and dehulling can be used, as well as expanders and collet dryers/coolers. After the preparation process, the matrices are fed into the extraction system. The general process of oilseed processing is shown in Figure 2.

Once the seed arrives at the processing plant, it is usually weighed and then sent to the cleaning process. These two processes are generally similar to those in the storage plants (described above). The traditional process continues with cracking; a series of two or three wavy high corrugated rollers rotate at relatively high speed, breaking the grain into several pieces. Rollers can have different configurations: pairs of horizontal corrugated rollers that rotate, rollers that rotate in a cylinder or pairs of rollers cavitated instead of toothed. The work capacity depends on the size of the machine but also varies from seed to seed. Modern cracking mills can process up to 1000 tonnes per day of oleaginous materials each [5]. In addition, other technologies can be used for seed crushing, usually during the dehulling process. These include hammer mills and disk attrition mills, where the seed is fed through a hopper into the centre of vertical, corrugated disks. From here, the particles are thrown outwards, where they are collected. Another technique is based on pneumatic impact, where the seeds are blown against a wall, causing them to break [8].

Many oilseeds are dehulled during preparation. Extraction without dehulling is possible, but the hull usually contains no fats and thus reduces the capacity of the plant. In addition, the hulls may contain components that should later be removed from the oil (e.g., waxes, pigments) so as not to reduce its quality [7]. Even before screw pressing, it is better to remove the hulls because they absorb some of the oil released by the grains. In addition, the hulls are rich in fibres and are more abrasive, leading to a higher wear of the press and horsepower consumption [12]. The proportion of the hull in the total weight of the seeds varies greatly, e.g., it is 7% for soybeans, 15% for rapeseed and 30% for sunflower seeds. Rapeseeds are usually not dehulled, while the hulls of sunflower and soybean seeds are usually removed. Although the proportion of hulls in soybeans is low, they are very often dehulled to obtain a protein-rich meal after extraction [8]. The dehulling is essential when the final product is not an isolated protein but a concentrated one. In other words, when proteins are extracted for further purification (during isolated protein production), it is not essential to start from a previously dehulled material, as the hulls contain few soluble substances that can cause problems during purification [13]. Various methods are used to separate the hulls from the kernels, including screening with vibrating sieves, air separation and electro-separation. Recent air separation technologies involve the use of cylindrical aspirators with multiple cascades. In these systems, the matrix is continuously fed to the upper part of the aspirator. In successive separation cascades, the material is repeatedly exposed to a transverse airflow that blows the light particles (hulls) to the centre, while the heavy particles (kernels) are directed to the product outlet. Figure 3 shows the usual soybean dehulling process, in which the hulls are separated from the kernel with a stream of air and collected in a cyclone or bag philtre. Since some kernel particles are sucked in with the husks, it is customary to carry out a second separation of the hull’s fraction. In electro-separation, the mixture of hulls and kernels is passed from a vibrating chute onto a roller connected to a corona electrode. The electric field generated by the corona electrode causes the hulls and kernels to be deflected differently; the hulls are deflected more than the kernels and are therefore collected in different boxes [8].

Hot dehulling (dehulling the seed while it is still hot) is widely used today, especially in the processing of soybeans. This results in energy savings and in a lower production of fine particles compared to the conventional system, using cold grain. Common hot dehulling systems consist of drying the grain from storage moisture to process moisture, dehulling the seed while it is still hot and delivering the conditioned cracks to the flakers without an intermediate cooling phase. In addition, these systems reduce residual oil content and refining losses.

Due to the difficulties associated with efficient dehulling before extraction, post-extraction dehulling methods have been developed. Solutions have been proposed for both rapeseed and sunflower. In some systems, hydrocyclones are used to separate a fibre and a protein fraction. In this way, Sosulski and Zadernowski [14] obtained a rapeseed meal with a protein content of 45% after solvent extraction, recovering the 66% of the initial meal. McCurdy and March [15], also using solvent-extracted rapeseed, washed the meal with water at pH 4.5 (9:1 *w*/*w* water/meal, 22 °C). After drying, the meal was finely ground and sieved through a vibrating sieve (325 mm). In this way, they achieved an increase in the protein content of the meal from 40.4% to 46.7%, but the yield of the protein-rich fraction was relatively low (39% of the treated mass). To obtain meals with a higher protein concentration, Murru and Lera Calvo [16] described a process combining milling, air classification and gravity separation. Starting from a dehulled industrial meal (36.5% protein), this system yielded a fraction with 43% protein, corresponding to 65% of the initial flow. Currently, this process is used in the industry for the production of meal with a high protein content [13].

After cracking and/or dehulling, the seed is processed in a conditioner (or cooker) where gentle heat is applied to soften the seeds (heated to a temperature between 60 and 75 °C) and make them malleable for subsequent fine reduction and/or flaking. Apart from cold pressing, all oil extraction processes require heating and sometimes further drying of the oilseeds before extraction. There are different types of equipment for heating/drying oilseeds. The conditioning is important to achieve a high oil yield and to give the seed the right elasticity, preventing the flakes from finely crumbling at low humidity. The most common are a drum conditioner, stack conditioner, plugged flow conditioner and hot air conditioner [5]. The stack conditioner (vertical), usually coupled with screw presses, and drum conditioner (horizontal) are the most common. These first ones consist of cylindrical vessels with a multitude of horizontal trays. The heat to raise the particle temperature and evaporate the moisture is conducted into the oleaginous materials from the upper surface of the trays filled with high-pressure steam.

The stack conditioner requires limited floor space but has the disadvantage that the residence time of the particles varies greatly. Drum conditioners consist of a series of horizontal drums (steam jacketed) that can be heated with steam up to 10 bar. The machine is slightly inclined, with the outlet end lower than the inlet end, so that the material advances each time it goes down. The rotation of the pipes raises the material continuously and it is discharged after about a third to half a revolution. They offer more consistent quality as the controlled seed flow provides more even heating and residence time [8].

Generally, oilseeds are flaked before solvent extraction. A flaking mill has two large diameter rollers that rotate in opposite directions and are forced together by hydraulic cylinders. As the seeds are pulled through the flaking mill, they are stretched and flattened. The flaking process helps to ensure more uniform cooking prior to screw pressing and allows more efficient extraction of the oil in the solvent extraction plant [12]. The shape of the flakes, with their large surface-to-volume ratio and the small distance between the oil cells and the flake surface, favours contact between the solvent and the solid and the migration of the oil into the solvent. However, flaking releases moisture that migrates to the flake surface and can hinder solvent penetration into the material. To overcome this problem, the surface moisture is removed by passing a strong stream of air through the discharge hoppers and conveyor belt of the flaker. Depending on the amount of seeds processed, there are different sized rollers (70–157 cm or larger) that press the seeds into flakes about 0.20–0.37 mm thick [5]. The characteristics of the material for good flaking are a moisture content of about 10% and a temperature of about 70 °C.

Before mechanical screw pressing, the flakes can be cooked in systems similar to those used for conditioning. Oilseeds are cooked or tempered to denature proteins, release oil from cells and inactivate enzymes. Cooking is conducted at high humidity, usually 9–12%, and the flakes are then dried to bring the humidity down to 2–5%. This is performed in ventilated vessels, usually with a stream of air flowing through them, and with a heat source to raise the temperature of the flakes to 104–120 °C [12].

Another process widely adopted by oilseed processing plants is the use of expanders. The expander consists of a short heat treatment at high vapour pressure followed by a rapid pressure drop to atmospheric pressure [17]. Expanders are mainly used for soybeans and cottonseeds after conditioning and flaking of the matrix. After the expansion process, the density and porosity of the matrix increase significantly, and the seed proteins are denatured. These effects deconstruct the seed and increase the mass transfer during oil extraction. Extrusion and the expander differ mainly in the source of heating energy. While the expander heats the seed by saturated and superheated steam injection, the heating in extrusion derives from mechanical sharing between the screw and the container wall. The implementation of expanders in oilseed preparation systems increases the bulk density of the matrix entering the extractor, reduces the solvent retention of the meal in the desolventiser, improves percolation in the extractor, reduces the levels of non-hydratable phospholipids in the degummed oil and improves the qualitative–quantitative production of lecithin (increase in lecithin recovery by 50–100%) [18]. The high pressure reduces the required residence time by a factor of sixty to a residence time of 30 s [19]. The short residence time allows the use of a much smaller and less expensive vessel and leads to a better quality of the oils and meal. The system is, however, limited in its application to products with a low oil content. Seeds with a high oil content (>40%), such as rapeseed and sunflower seeds, do not usually allow sufficient pressure build-up in the units to achieve similar effects as seen with soybeans [20].

Expanders work in the same way as mechanical screw presses but with a lower pressure. The system consists of a horizontal cylinder fed by a screw that transports the material. The cylinder can be heated and is endowed with nozzles for injecting water and live steam to increase the temperature, humidity and pressure inside the cylinder. In the first section of the system, the seed is brought to a water content of 10–15% and heated to 105–120 °C. Subsequently, the mechanical pressure in the extrusion phase leads to a further increase in temperature (about 160 °C). The material is forced through the outlet, which can be a plate with several openings or a hydraulic cone. When the product leaves the expander and is released to the ambient pressure, there is rapid evaporation of the water and expansion of the product, which takes on a sponge-like consistency. There are different types of expanders: closed-wall and slotted-wall expanders. Unlike closed-wall expanders, slotted-wall expanders allow the controlled release of excess oil through the slotted wall and produce collets with 20–30% oil. With this configuration, it is possible to process non-pre-pressed materials with a high oil content (e.g., full-fat sunflower), producing collets ready for solvent extraction [12]. On leaving the expander, the hot, moist matrix is often cooled (to around 60 °C) and dried (2% moisture reduction) before extraction. Direct air cooling is the most common method for mass cooling after expander treatment. Dryers also allow the matrix to reach a more uniform humidity and temperature. During extraction, an uneven matrix can result in a miscella that is more prone to foaming, coating or clogging the evaporator tubes during second effect evaporation.

Alternative techniques for cooling the collets have been proposed to reduce energy consumption. These include the use of carbon dioxide (CO_2_) snow, which is produced when liquid CO_2_ is released at high pressure through a special spray nozzle at atmospheric pressure. The possibility of using CO_2_ snow to cool the collets of soybean expansions was discovered about 25 years ago [19]. However, this method proved to be not cost-effective, as a cryogenic reservoir of liquid carbon dioxide must be available on site and the CO_2_ often has to be sourced from distant producers. In contrast, the use of water indirect heat exchangers proved more efficient, consuming up to 90% less energy than other technologies. These systems consist of one or more rows of vertical, hollow, stainless-steel plates in which the product flows slowly by gravity (residence times of 5–10 min). The cooling water flows through the plates in countercurrent to the flour tongs and cooling is based entirely on conduction. The system is usually connected at the bottom to an extraction screw press that generates a mass flow and regulates the flow rate. This process results in remarkably stable and uniform final product temperatures. As the air does not encounter the product, the risk of bacterial contamination and a change in the moisture content of the product is avoided. Finally, the water is used repeatedly in a closed loop system [19].

For grains with low oil content, such as soybeans, the flakes are usually delivered directly to the solvent extraction plant. For oilseeds with higher oil content, such as sunflower or rapeseed, or for virgin oils (extracted mechanically only), the flakes are usually delivered to a mechanical pressing plant. For complete oil extraction, after mechanical pressure, cake is usually extracted with solvent.

### 3.1. New Sampling Techniques for Heat-Damaged Imported Seeds

Brazil, the United States and Argentina are among the main producing and exporting countries of oilseeds, which are often exported overseas. China is the world’s largest importer of soybean. Oilseeds are therefore often transported over long periods of time and can be damaged by high storage temperatures or poor ventilation during transport. Heat damage directly affects the quality of the product and significantly reduces the value of the seeds. In soybeans, heat damage negatively affects the properties of the protein isolate (protein denaturation) and the oil (reduced polyunsaturated fatty acid content and increased *n*-hexaldehyde content). Assessing the quality of imported oilseeds is therefore essential. The current method for identifying heat-damaged oilseeds is manual sorting based on colour differences. However, this form of inspection is time-consuming, and the results are inevitably subjective. Furthermore, denatured proteins associated with heat damage do not always cause discolouration of the oilseeds. To improve the accuracy of identification, mechanised control has been introduced, as a fast, accurate, objective and reproducible method. However, this automated vision cannot provide detailed information on chemical composition because it is limited to the visible range. To overcome this problem, Near Infrared Spectroscopy (NIR), which has emerged as a non-destructive identification technique for studying the physical and chemical properties of materials, can be exploited. Indeed, changes in oilseeds affect light scattering and energy absorption patterns, resulting in differences in NIR absorption between heat-damaged and healthy oilseeds [21,22]. In addition to NIR spectroscopy, the emerging hyperspectral imaging (HSI) technology offers the advantages of rapid and non-destructive analysis. HSI also integrates spectral information with spatial information from imaging techniques. Furthermore, HSI requires no sample preparation and scans many samples simultaneously. A recent study investigated the possibility of identifying heat-damaged soybeans using HSI. The results show that the HSI technology is an accurate, effective and non-destructive technique for classifying sound- and heat-damaged soybeans [23].

### 3.2. Microwave, Ultrasound and Pulsed Electric Fields Pre-Treatments

Recently, Koubaa et al. (2016) described the use of new technologies such as ultrasound (US) and MW to improve the yield and quality of oil extraction from oil crops. In the extraction of plant matrices, a partially damaged cell wall significantly improves the accessibility of the oil for extraction. In all the reported articles, US was used during extraction with various solvents (e.g., hexane, isopropanol, supercritical CO_2_) and not as a pre-treatment of the matrix [24]. US produces acoustic cavitation, which consists of the formation, growth and decay of gaseous bubbles in a liquid. The cavitation effect on the matrix improves the extraction yield through the mechanisms of erosion, fragmentation, sonoporation and the so-called ultrasonic capillary effect [25]. Zdanowska et al. [26] evaluated the effects of pre-treatment with ultrasound on the process of the continuous oil pressing of rapeseed. Pre-treatment with US did not significantly influence the oil yield, but the temperature during pressing was much lower and the flow rate of the oil was higher over time. In summary, the energy efficiency of the press was about 25% higher when the seeds were pressed after pre-treatment with US. However, some unfavourable changes in the oxidation stability of the oil were observed after US pre-treatment, and the need for additional drying of the seeds before pressing could be economically disadvantageous. As far as the authors are aware, the industrial use of US to pre-treat the matrix before oil extraction has only been used in the production of extra-virgin olive oil (Clodoveo et al. [27]). The use of a continuous flow sono-heat exchanger, which combines the mechanical energy of US with the ability to modulate the heat exchange of the olive paste, eliminated malaxation, increased the extraction yield and improved the polyphenol content of the oil.

Microwave (MW) irradiation is an energy-efficient alternative to conventional heating treatments. Among the various new methods available, MW-assisted pre-treatment of oilseeds has proven an efficient method for producing high quality vegetable oil with high nutritional aspects. Furthermore, it can increase oil extraction yields compared to conventional extraction under the same conditions. MW pre-treatment of oilseeds increases mass transfer as it favours the rupture of the cell membrane. In addition, the formation of pores in the cell walls makes them more permeable to the passage of oil during extraction. An example of a continuous industrial-scale plant for the pre-treatment of oilseeds with microwaves was proposed by Koubaa et al. [24]. In such a plant, the seeds are transported by a screw conveyor along horizontal pipes coupled with MW horns, into which steam is injected (Figure 4). In addition to the higher yield and high oil quality, MW treatment allowed lower energy consumption, faster processing times and lower solvent consumption compared to conventional methods. However, high MW power or excessively long pre-treatment times can have a negative effect on oil quality, e.g., by reducing the amount of polyunsaturated fatty acids [28].

Azadmard-Damirchi et al. [29] showed that a short MW pre-treatment of rapeseed (2–4 min) resulted in an increase by 10% in the oil yield extracted with the press. In addition, the pre-treated oil was richer in phytosterols (15% increase), tocopherols (55% increase), canolol and phenolic compounds, resulting in higher oxidative stability compared to the oil extracted without pre-treatment. Zhou et al. [30] investigated the influence of MW pre-treatment on the flavour properties of rapeseed oil extracted by cold pressing. The profile of the volatile components of rapeseed oil was positively affected by MW pre-treatment. A treatment at 800 W for 6 min was indeed sufficient to improve the flavour of rapeseed oils by releasing the pyrazine compounds that impart a pleasant roasted flavour to the oil. The authors concluded that the use of this technology as a pre-treatment process would allow the production of pressed oils with a special flavour [30]. Despite the advantages presented, MW pre-treatment of oilseeds in large-scale plants has not yet been reported. There is also a lack of information on the economic costs of this process in large-scale production.

Due to the electroporation of the cell membrane, pulsed electric fields (PEF) are used as an innovative pre-treatment to facilitate the recovery of intracellular material, such as oil, in the subsequent extraction step. Pre-treatment of rapeseed and sunflower seeds with PEF has proved to increase oil yield in both mechanical and solvent pressing extraction. A continuous flow pre-treatment and/or extraction system under combined US and PEF conditions is shown in Figure 5 [25]. Guderjan et al. [31] showed that pre-treatment with PEF increases the rapeseed oil extraction yield and the concentrations of total antioxidants, tocopherols, polyphenols and phytosterols in the oil. Moradi et al. [32] evaluated the effects of pre-treatment with US and PEF both individually and in combination on the extraction of sunflower oil with hexane. The results showed the superiority of PEF treatment over other methods in terms of oil yield. Scanning electron microscopy showed that the surfaces of the PEF-treated samples were more porous, and thus the solvent diffusion rate may be much higher. However, PEFs require an aqueous medium to be effective, so the matrix need to be soaked or directly immersed in water before treatment. This requires an additional, energy-consuming drying step of seeds after pre-treatment. In addition, both studies described lab-scale systems, so further investigations are needed on the application of this technology on a pilot scale.

### 3.3. Instant Controlled Pressure Drop Technology

Instantaneous controlled pressure drop (DIC) technology is a thermomechanical process that uses a short-term treatment at high temperature and pressure. In DIC, biological matrices are subjected to a saturated vapour pressure of 100 to 900 kPa for a few seconds, followed by an abrupt controlled pressure drop at a rate of more than 500 kPa per second, leading to an absolute ultimate vacuum of 10 to 5 kPa. The instantaneous pressure drop leads to instantaneous autovaporisation of the water, rapid cooling of the biological products and expansion and texturization of the matrix. DIC is widely used in the food industry, e.g., for microbial decontamination, deodorisation, swell drying and texturization [33].

DIC has also been investigated as a possible processing technique for oilseeds prior to oil extraction. The porous structure achieved by DIC improves mass transfer and increases both the effective diffusivity and accessibility of the matrix, thereby improving the overall kinetics of oil extraction [34]. DIC has a very high texturing capacity, which can lead to rupturing of the cell walls. The intensity of texturization usually depends on the amount of vapour produced by autovaporisation, which is directly related to the temperature drop. Oilseed treatment with expanders, which is usually carried out at an absolute vapour pressure of 0.6 MPa and a treatment temperature of about 160 °C, results in a temperature drop of 60 °C when decompressed to atmospheric pressure. Under similar treatment conditions, DIC flash decompression at a vacuum of 4 kPa (equilibrium end temperature of about 30 °C) results in a temperature drop of 130 °C. The texturing effect of DIC is therefore greater [17]. A comparison between DIC and expander treatments is shown in Figure 6.

Pech-Almeida et al. [33] provided an overview of the use of DIC in the food industry, including its use in the extraction of oilseeds. Several studies investigated the effects of DIC on the extraction of oilseeds (such as rapeseed, soybean and sunflower) by mechanical and solvent extraction. In the most recent paper, Jablaoui et al. [17] compared the effects of expanders and DIC on the extraction of soybean oil with *n*-hexane. The use of DIC resulted in higher extraction yields with faster extraction kinetics. To achieve the same yield as that obtained in 160 min for cracked or cracked/flaked soybeans, the expander required 120 min, compared to 35 min for DIC. Due to the brevity of the heat treatment and immediate cooling, DIC maximally preserved the quality of the soybean oil, with fatty acid concentrations almost identical to those of the untreated seeds. The authors concluded that pre-treatment with DIC can substitute expanders by reducing the extraction time and increasing yields, while maintaining oil quality. However, the large quantities of treated seeds in the food industry lead to questions regarding the feasibility of this technology. A flow-through concept for DIC is still a long way off today and further research in this direction is needed.

An alternative pre-treatment method to intensify the extraction process is to bring the natural material into contact with CO_2_ under high pressure and then decompress it under atmospheric conditions. This pre-treatment showed a positive effect on the extraction kinetics, reduced solvent consumption and, in some cases, improved the extraction yield. The efficiency of pre-treatment with CO_2_ depends on the exposure time to the dense gas atmosphere, the pre- and post-expansion pressure, and the decompression rate [35]. Meyer et al. [35] studied the effect of rapid CO_2_ decompression pre-treatment on rapeseed and sunflower seeds. A significant positive effect on extraction directly related to the pre-treatment was evident for sunflower seeds but not for rapeseed. The authors concluded that the mechanism controlling the process is the macroscopic destruction of the material. Although it cannot completely replace conventional mechanical treatments, rapid CO_2_ decompression could be applied in addition to those to possibly avoid the frictional heat stress of extensive mechanical pre-treatment methods.

## 4. Oilseed Extraction

### 4.1. Oilseed Mechanical Extraction: General Process and Parameters

Due to the relatively low oil yield obtained by mechanical extraction, this technique on its own is used far less frequently than solvent extraction or the combination of both methods (mechanical followed by solvent extraction). Mechanical extraction can reduce the oil content in the meals to 5–10% by weight, whereas solvent extraction reduces the oil content to less than 1%. From an economic point of view, the value of the oil fraction is usually two to three times higher than the weight value of the meal fraction. In addition, the energy and maintenance costs for the mechanical extraction process are relatively higher compared to solvent extraction, per tonne of oil processed [5]. Nevertheless, the mechanical extraction is preferred for certain productions. It is indeed cost-effective for very small ones (about 10 tonnes per day) as the capital costs are much lower than for small solvent extraction plants. Moreover, there is a high-value niche market for natural oils and flours that have not been in contact with solvents and are obtained exclusively by mechanical extraction. Mechanically pressed oils are generally suitable for direct consumption and do not need to be refined. A general screw expression process is presented in Figure 7.

A typical mechanical pressing process involves cooking, pressing, cake cooling and oil filtration. As with most of the processes described in this review, there are many variations in mechanical pressing plant design. There are several purely mechanical extraction methods: cold pressing, pressing after cooking and double pressing (cold pressing followed by heat conditioning and a second pressing). Their performance in terms of residual oil content in the meal ranges from 9.9 to 16.3%, 9.7 to 15.7% and 8.7 to 13.5% residual oil (rapeseed), respectively [36]. The key to the press performance is the conversion of electrical energy into pressure rather than heat.

The pressing process is generally divided into pre-pressing or full pressing of the oilseeds. Pre-pressing yields about 60% of the available oil, while full pressing affords almost 90%. Oily materials with more than 30% oil by weight tend to decompose in the extractor after most of the oil has been extracted, resulting in poor final extraction and high solvent retention. Solvent extraction is most effective when the oily material contains about 20% of oil. In this condition, the system is in thermodynamic equilibrium, as the waste heat from desolventising of the meal is sufficient to serve as the main heat source for evaporation of the solvent in the miscella. Therefore, oily materials with an oil content of more than 30% (e.g., rapeseed/canola seeds, sunflower) are usually reduced to 20% by pre-pressing prior to solvent extraction.

Oilseeds that are pre-pressed prior to solvent extraction are usually heated to a temperature between 75 and 110 °C to reduce the viscosity of the oil and to obtain a good quality cake. The heating is often performed in two stages: the first to about 65 °C before flaking and the last one to about 100 °C before entering the press [10]. Another type of pressing, known as cold pressing, has become popular in recent years. In cold pressing, the product is not heat-treated (pressing temperature < 60 °C) and high-quality oils are produced (e.g., reduced phospholipids content). However, the oil yield is considerably lower [37].

A pre-press is a mechanical device that uses a horizontal screw of increasing diameter to apply pressure to the oily material as it moves along the screw. The diameter of the cake outlet, the arrangement of the throttle and the design of the screw determine the pressure inside the press. The barrel surrounding the screw is grooved longitudinally so that the increasing internal pressure first expels the air and then some of the oil through the barrel. The screw turns at a relatively slow speed (in the order of 10 to 60 revolutions per minute). The cylindrical drainage cage consists of two longitudinal halves that rest against the worm shaft and whose sections are screwed tightly together to withstand the pressure. The worm shafts are configured so that the compaction of the oilseed increases progressively as it is pushed from one segment to another. This ensures that compression does not decrease as the volume of the oilseed decreases due to the compression of solids and the escape of non-compressible oil. This is performed by reducing the depth of the channel (the open space between the inner diameter of the cylinder and the surface of the hub of the central shaft) or reducing the pitch of the successive worm flights. Sometimes both techniques are used [12]. The geometry of the pressure surfaces should be designed to limit slippage and improve axial pressure transfer to achieve higher pressures and avoid unnecessary heating of the solids [36]. In general, during pre-pressing, the presses reach 30–40 bar and a temperature of about 95 °C. Modern pre-presses are capable of processing 500 to 1000 tonnes of oily material per day. The ratio of the available volumes between the press inlet and outlet gives the compression ratio of the press. Ideally, this ratio should be between 4 and 4.5 for seeds with high oil content (50%). In industrial presses, this ratio is usually in the order of 10 to compensate for material slippage along the screw [38].

The equipment used for full pressing is similar to pre-press systems but generally operates at higher temperatures and pressures. The secret to the performance of a full press is to apply maximum pressure to a thin section of oily material to squeeze out as much oil as possible. Because full presses generate an intense heat, the shafts are often water- or oil-cooled to dissipate the heat and maintain adequate internal friction and pressure. During full pressing, pressures of about 400 bar and temperatures of 115–125 °C are generally reached, drying oilseeds to 3% moisture. The high temperature reduces the viscosity of the oil, making it easier to be completely pressed out. In addition, the high degree of drying breaks down the cell structure of the oily material as the internal moisture evaporates causing an expansion. The low final moisture content maximises friction in the press. All these aspects help to minimise the residual oil content in the fully pressed cake. Currently, high-capacity full presses produce residual oil levels of 5–8% [10]. Most full presses can process 10 to 100 tonnes of oily material per day. Savoire et al. [38] reported on the capacity, energy consumption and extraction efficiency of several commercially available industrial presses. The largest capacity solvent-free crushing units in the world process more than 250,000 tonnes but are mainly related to biodiesel production [36]. For these reasons, the new requirements for screw presses favour a higher capacity, lower horsepower consumption and easier maintenance. Modern plants can process about 800 tonnes per day with installed motors of up to 630 kW, and manufacturers already have plans for even more powerful machines with drive sizes close to 1 MW [20].

Savoire et al. [38] summarised the effects of operating parameters (screw speed, temperature and back pressure) and raw material (seed type, variety, water content and pre-treatment) on the performance of the mechanical pressing process (oil yield and pressing capacity). However, due to the interdependence between these parameters, it is relatively difficult to assess the effect of each parameter independently of the others. An overview of the effect of the individual parameters is shown in Table 2.

The oil does not exit the area of higher pressure but must flow back to find an outlet. The goal is to allow the oil to travel as short a distance as possible from the high-pressure area to the outlet. The ejected oil is collected in a basin below the screw and the partially de-oiled cake emerges at the end of the screw. Pressing has two important functions: to partially de-oil the material and to produce a porous cake with sufficient structural integrity to allow high efficiency in downstream solvent extraction [5].

The oil obtained by mechanical pressing usually contains a high concentration of meal fines (about 5–10% by weight), which are removed in a screen kettle and then in a leaf or plate filter before the oil is sent to the refining process. In the traditional method, the oil is pumped into a tank where a residence time of 30–60 min is observed to allow the heavier particles to settle and be removed from the bottom of the tank. After gravity separation, the oil is then pumped under pressure through a leaf filter to finally separate the fine particles. Alternatively, a high-speed centrifugal decanter is used to separate the fine particles from the oil. The separated fine fraction, rich in oil, is generally recycled back into the process at the inlet of the press. A reduction in fines can be achieved by using expanders before pressing or in some cases after pressing to agglomerate the fines before the solvent extraction. After the centrifugal decanter, the prepress oil is typically in the range of 0.1% solids content and 0.2% moisture content. If the pre-press oil has to be degummed, it is generally mixed with the solvent-extracted oil and then degummed together, whereas if it needs to be stored before further processing, it is generally passed through a vacuum dryer to reduce the moisture content below 0.1% and through a cooler to reduce the temperature below 50 °C before being pumped for storage [39].

During pressing, the cake passes through the end plate where it is compressed by the high friction and becomes quite hard. The cake is usually crushed and cooled before being fed to the solvent extractor (pre-pressing) or used directly as animal feed or fertilizer (full pressing). However, it is considered to be a hazardous cargo that may self-heat due to high moisture, residual oil or both. The self-heating process, although slow, may cause the temperature of the cargo mass to rise to the point of spontaneous combustion. The shipper must provide a certificate issued by a person recognised by the competent authority of the country of dispatch, confirming the oil and moisture content of the cargo. The International Maritime Solid Bulk Cargoes (IMSBC) code classifies five types of cakes with specific loading and carriage requirements, depending on their oil and moisture content and method of extraction [40]. Press cakes are generally more hazardous than solvent-extracted meals because they have a higher residual oil content and a moisture content of between 7.3 and 11.9% [36].

#### 4.1.1. Oilseed Mechanical Extraction Optimization

New types of presses have been introduced to the market, which allow direct pressing after seed cleaning without further pre-treatment of the seed. This eliminates the need for breaking, dehulling and conditioning of the seeds. These machines use throttles, thus imitating to some extent the expander systems. However, these presses are 60% longer, 70% heavier and more expensive. The power required to drive them is about twice that of conventional presses. However, such systems are highly efficient (<2% residual oil content) and their adoption results in energy savings due to the elimination of pre-treatment steps. Some manufacturers state a 25% lower energy consumption for the entire process [8].

Optimising energy consumption and reducing the environmental impact of the whole system are the main goals of industrial plants today. This can be achieved by using fewer and more efficient motors and by working with better prepared seed (e.g., by coupling the use of expanders immediately before pressing). A lower environmental impact would be achieved by avoiding the escape of oily mists and pungent fumes. Oily vapours easily escape through the jacket surrounding the cylinders and enter the pressing room, causing environmental problems in the processing plant and in the airflow from the plant. Some presses have been fitted with ducts on the cover that allow airflow to draw in the vapours and direct them to a deodorising scrubber that reduces the odours escaping into the atmosphere [19]. To reduce energy consumption, some screw presses have been designed to eliminate the need for forced feeding. These screw presses are fed by free-falling influent streams discharged by variable speed screw conveyors. Moreover, some of the screw presses available today are supplied with adjustable chokes. An adjustable choke allows a minimum of residual oil to remain in the cake, even if the screws wear out. Without an adjustment system, the pressure exerted by the shaft would steadily decrease as the screws wore, and this pressure drop would lead to a gradual increase in residual oil. The screw shafts of most screw presses can be provided with internal bores for water cooling. Some manufacturers also offer water-cooled cages or an oil-cooled washing system to cool the cages and wash off solids that adhere to the cages. Furthermore, some screw presses allow the number of revolutions of the screw shaft to be adjusted during pressing. Changing the speed of the screw shaft affects the compression exerted by the screw shaft and is a valuable means of optimising pressing performance [19].

#### 4.1.2. Twin-Screw Extrusion Technology

Twin-screw extrusion is an innovative technology that is increasingly used in the polymer, grain, pet food and paper industries and, more recently, for the mechanical extraction of oilseeds. Unlike single-screw, twin-screw extrusion technology is not currently used on an industrial scale for the mechanical pressing of oilseeds.

In single-screw presses, friction is the main factor, resulting in high energy consumption, which, combined with poor mixing capacity, can lead to overheating and the consequent deterioration of oil and cake quality. Twin-screw extrusion, on the other hand, works the same way as a positive displacement pump, which means that the productivity of the process is independent of the pressure profile and screw speed. Furthermore, with twin-screw extrusion, the possibility of introducing mixing blocks leads to much better mixing, resulting in better heat transfer and product uniformity [41].

Twin-screw extruders consist of two intermeshing co-rotating or counter-rotating screws mounted on grooved shafts and enclosed by a modular barrel. A filtration module collects the liquid filtrate, which is expelled from the material by compression. Various designs, such as intermittent and co-rotating screw systems, ensure efficient mixing and heat transfer. A design with intermittent counter-rotating screws results in longer residence times and high pressure and shear in the upper intermittent screw zone, leading to intensive mechanical processing of the material. The high variability in the twin-screw extruder’s configuration allows the pressure profile within the extruder to be precisely adjusted to optimise oil extraction efficiency.

Uitterhaegen and Evon [41] provided an overview of the state of the art in the use of this technology for vegetable oil extraction. Twin-screw extrusion allows for significantly higher mechanical energy input, leading to energy savings up to 80%, compared to single-screw extrusion. In addition, the high flexibility of twin-screw extrusion allows different oilseeds to be processed with minor equipment changes, whereas single-screw presses are often designed specifically for a single type of oilseed. The possibility of the thermomechanical treatment of oilseeds at different stages of operation could eliminate the need for seed pre-treatment steps. With twin-screw extrusion, several independent process variables can be set, including feed speed, screw speed and temperature profile along the screw axis. This leads to high process flexibility and optimisation potential.

The implementation of a twin-screw extruder as a continuous solvent extruder was proposed as an innovative extraction process. Modified twin-screw extrusion processes with solvent injection pumps were developed and evaluated. Solvents tested included water, alcohols acidified with phosphoric acid and fatty acid methyl esters (FAME). Among these, FAME proved to be the most efficient extraction solvent, allowing the recovery of up to 98% of the total oil (based on the residual oil content in the meal). As FAME has a higher oil extraction capacity, lower liquid to solid ratios can be used and a better oil separation is achieved compared to water extraction [41].

In another oil extraction process, a twin-screw extruder with two separate filtration modules was used to obtain a first filtrate consisting of the pressed oil from the pressing stage and a second filtrate downstream of the solvent extraction stage. These combined processes have great industrial potential due to their efficiency and flexibility [42]. A schematic representation of the twin-screw extruder combining thermomechanical extrusion and solvent extraction is shown in Figure 8.

Overall, significant improvements in oil extraction and quality have been achieved using twin-screw extruders. However, this technology is still at an early stage of development and further research is needed to improve its overall performance, validate the efficacy for various crop species and assess the industrial feasibility.

#### 4.1.3. Gas-Assisted Mechanical Expression

Two possible alternatives for oil extraction are supercritical carbon dioxide extraction (SFE-CO_2_) and a newer process called gas-assisted mechanical expression (GAME). In contrast to SFE-CO_2_, in the GAME process the CO_2_ is dissolved in the oil contained in the seeds before pressing. The mixture of oil and CO_2_ is then pressed in a hydraulic press or a screw press. The liquid in the GAME press cake is saturated with CO_2_ (up to 30% *w*/*w* of CO_2_). After pressing, the CO_2_ can easily be removed from the cake and oil by depressurisation. When the cake is depressurised, some of the extra oil is removed by entrainment in the gas stream, especially if the seeds are not dehulled before pressing. The resulting GAME cake has a lower residual oil content than that produced by conventional pressing under the same conditions. The oil yield can be up to 30% higher than with conventional pressing. Willems and de Haan [43] schematically described a two-stage extruder for GAME (Figure 9).

A pressure of 10 MPa is a good compromise to achieve increased yield with reduced working pressures. At this pressure, the viscosity is about an order of magnitude lower than that of pure oil. According to the Darcy equation, pressing CO_2_-saturated oilseeds should be 10 times faster.

The advantages of GAME have been summarized by Willems and de Haan [43], and include: (i) a mechanical pressure lower than conventional pressing (10–50 MPa compared to 100 MPa) while still affording higher yields; (ii) the amount of CO_2_ required is about ten times lower than for SFE-CO_2_; (iii) the extracted oil is completely free of toxic solvents; and (iv) a sterilising effect of CO_2_ under the conditions used, as several authors pointed out.

The higher oil yield obtained with GAME is due to the lower viscosity of the oil–CO_2_ mixture compared to pure oil, the degradation of cell walls during depressurisation and the increase in volume of the liquid content (oil compared to CO_2_-saturated oil). On an industrial scale, continuous operation is required to make the process economically feasible. For a continuous flow GAME, the use of a two-stage extruder was proposed. In the first stage, part of the oil is removed by mechanical pressure alone. After that, the oilseed paste can be expanded and actively mixed with CO_2_ before being pressed a second time. The operation of GAME in a two-stage continuous extruder seems feasible.

Müller and Eggers [44] investigated the efficiency of industrial scale GAME on rapeseed. The residual oil content in the cake was reduced to 7.7% (*w*/*w*) by GAME at an average CO_2_ pressure of 12.5 MPa. Conventional pressing with the same process resulted in an oil residue of 9.9%. In addition, the oil quality was improved and the payback time of the investment was calculated to be about 3 years under these conditions.

In summary, GAME is a promising technique for increasing the oil yield of oilseeds without the use of organic solvents. Thus far, GAME has mainly been investigated in single-axis hydraulic presses. Further research is needed to optimise the GAME process in continuous flow extruders. This technology was brought to the industrial stage by the company Harburg-Freudenberger (Hamburg, Germany) in connection with Crown Iron Works Co. (Minneapolis, MN, USA) under the name Hiplex [45].

### 4.2. Oilseed Solvent Extraction

#### 4.2.1. General Process and Parameters

The solvent extraction process consists of five closely related subprocesses: solvent extraction, meal desolventisation, drying and cooling of the meal, distillation of the miscella and solvent recovery. Extraction is always used when a residual oil content of <2% is desired. Most vegetable oils are obtained by solvent extraction.

The solvent extraction process has the advantage of a much higher oil yield (<1% residual oil in the meal) than mechanical extraction, as well as lower operating costs per unit. However, solvent extraction plants have high initial costs, not least because the entire plant must be explosion-proof. Typically, these plants process between 1000 and 5000 tonnes of seed per day, with construction costs ranging from USD 15 million to USD 75 million.

Ideally, solvents used for the extraction of vegetable oils should show high oil solubility at low temperatures, high selectivity for lipid compounds, chemical inertness (non-explosive), non-toxicity, low viscosity and surface tension, low boiling point and low heat of vaporisation, and they should be environmentally friendly. Obviously, no extraction solvent can fulfil all the properties from the above list. Therefore, the best compromise must be found based on priorities. Before hexane, carbon disulphide, benzene and trichloroethylene were used, which were later banned because of their high toxicity [5].

The solvent exploited in most oilseed extraction plants around the world is hexane, a mixture of saturated acyclic hydrocarbons with six carbon atoms that distils between 64 °C and 70 °C [46]. Commercially available hexane contains about 65% of *n*-hexane. As well as this main component, hexane can also contain other hexane isomers, cyclic hydrocarbons such as cyclohexane or even aromatics such as toluene or benzene [47]. The low viscosity of hexane is favourable for extraction, especially for percolate extraction. The viscosity of hexane decreases almost linearly from 0.4 cP (0 °C) to 0.37 cP at 15 °C. Between 15 and 25 °C the curve is steeper and then returns to an almost linear curve up to 50 °C (0.25 cP).

The oil content in the mixture influences the vapour pressure and leads to different boiling points. Hexane vapours can form an explosive mixture with air, so special care must be taken when building and operating solvent extraction plants. Hexane/air mixtures are explosive in the range of 1.2 to 7.4% *v*/*v* hexane. Hexane extraction allows high oil recovery at low production costs. In addition, the meal defatted with hexane is odourless and has a low residual oil content, which makes it very marketable.

During solvent extraction, the miscella diffuses through the cell walls to the oil bodies inside the cells. As the miscella continues to penetrate and dissolve the oil, the pressure in the cell increases and the concentrated miscella diffuses back out of the cell. Once the more concentrated miscella reaches the miscella bath, the concentration of the bath gradually increases. This process continues until the miscella concentration in the cells of the oil material reaches equilibrium with the concentration of the miscella bath [3].

Six parameters affect the performance of solvent extraction systems: (1) contact time, (2) particle thickness, (3) number of extraction stages, (4), miscella flow rate, (5) extractor temperature and (6) solvent retention [5].

The extraction time needed depends on the kind of seed, its pre-treatment and the equipment used. Adequate contact time (1) is critical to maximise extraction efficiency. The residence time corresponds to the time the oily material spends in the extractor. The residence time can be divided into wash time and drain time. This latter is the time during which the oily material drains before being emptied, while the washing time includes the contact time and the dormant time. Extraction only takes place during the contact time, which is the time the oily material is in contact with the miscella. The dormant time is the time an oily particle spends in the wash zone of the extractor where it is not in contact with the miscella. The ratio between contact and dormant times varies depending on the extractor. Extractors with a deep bed of material and a small surface area generally operate by immersion and provide a very high contact to dormant times ratio. Extractors with a shallow bed depth and a large bed surface area generally work by percolation and offer a lower contact to dormant times ratio. Despite the higher initial investment, the long-term benefits of a longer contact time often justify oversizing the extractor. In the commercial operation of a shallow or deep bed extractor, the residence time for each extraction stage varies between 7 and 10 min or an average of 45 min for the entire extraction process.

The structure of the oilseeds (2) at the entrance of the extractor depends on the cell structure of the material, but especially on the preparation steps. For almost all oil materials, the preparation ends with flaking, which reduces its thickness and thus the distance and number of cell walls that the miscella must pass to reach the oil bodies. Depending on the structure of the oil material, the time needed for the miscella that enters the cells to reach equilibrium with the miscella outside changes. For example, soybean flakes have a cell structure that allows them to reach equilibrium in about 5 min each time the extractor is filled, while sunflower cakes take about 9 min and rapeseed cakes about 12 min. If all other extraction parameters remain constant, a smaller extractor can be used by reducing the particle thickness. However, a further reduction in particle thickness involves additional costs. For all oily materials, the economic balance between the initial cost of the extractor and the continuous electricity cost required for flaking can be analysed and the optimal particle thickness determined [3].

An extractor with only one miscella stage would require a large amount of solvent to achieve high extraction efficiency. The energy required to evaporate the solvent in the mixture would be enormous; therefore, multi-stage countercurrent extractors are used. By iterating the mass balance, the minimum number of stages in the mixture can be calculated for a given solvent/material ratio. For an energetically competitive distillation system, the solvent/material ratio must be kept close to or below 1. Theoretically, the higher the number of stages in the mixture, the higher the extraction efficiency. However, if the individual stages do not have enough contact time to reach equilibrium, the addition of stages will not further reduce the residual oil. In this case, more extraction stages simply mean more pumping and more potential solvent losses. The number of extraction stages (3) is generally determined by the total time of the wash zone and the number of stages that can theoretically reach equilibrium within the time of the wash zone. Most commercially available extractors deal with between five and nine extraction stages [3].

Miscella flow rate (4) is the maximum volumetric flow rate of miscella that can flow through the material bed per unit area of it (m^3^/h per m^2^ of material bed). The main causes of reduced miscella flow are the presence of thinner than normal flakes, high surface moisture or an abundance of fine particles. A uniform shape of the material arriving at the extractor ensures uniform miscella flow rates.

Increasing the temperature in the extractor (5) increases the diffusivity of the miscella through the cell walls of the oil material. To optimise extraction, it is therefore necessary to reach the highest possible temperature. However, the temperature influences not only the extraction rate but also the proportion of non-oil lipids (e.g., non-hydratable phospholipids) and non-lipid components in the crude oil [48]. The maximum operating temperature of the extractor depends on the safe working temperature of the solvent. Since the boiling range of commercial hexane at sea level is usually 64–70 °C, the highest possible temperature to avoid boiling is 63 °C. However, most extractors work at 60 °C to ensure a higher safety margin. To avoid heat loss, the oil transport system and the extractor are thermally insulated.

After the washing area of the extractor, the miscella retained by the extracted material is drained by gravity (6). This gravity drainage usually takes between 5 and 20 min and varies depending on the depth of the material bed in the extractor. The deeper the material bed, the longer the drainage time required. After drainage, meal from flaked oilseeds usually retains 30–40% by weight of solvent (hexane). Pre-pressed cake and oilseeds prepared by extrusion can free drain to 20–25% solvent before leaving the extraction vessel [12]. The more solvent retained in the meal, the more energy is required for the desolventisation step and the higher the risk of protein degradation. Solvents with higher polarity than hexane, especially in the presence of water, are better retained in the meal and are more difficult to be removed, resulting in higher energy consumption and a greater loss of protein solubility [49]. In the meal desolventisation system, the solvent is evaporated, leaving traces of oil, often referred to as residual oil. This contains high concentrations of phosphatides (about 20%) and other non-triglyceride compounds [5]. Adequate draining time of the extractor is the most economical way to minimise the retention of miscella.

To reduce operating costs, it is crucial to maximise the oil concentration in the miscella at the outlet of the extractor, reduce the retention of solvent in the wet meal and achieve the lowest effective solvent to solid ratio that results in acceptable residual oil in the meal [19]. Industrial solvent extraction plants are divided into batch extractors and continuous extractors. The latter follow the principle of countercurrent extraction and are divided into immersion extractors or percolation extractors. Some countercurrent extractors are shown in Figure 10.

Except for virgin oils, crude oils cannot be consumed directly or used in various foods without technological refining processes. Crude oils such as soybean, rapeseed, palm, corn and sunflower oils must be purified or refined before consumption. The aim of refining is to obtain an odourless and rather neutral-tasting oil that is limpid and colourless and free of contaminants [51]. Compounds known to have a negative impact on the quality and stability of oils include free fatty acids, unsaponifiables, waxes, pigments, solid impurities (especially fibres) and oxidation products (peroxides, aldehydes, ketones and oxidised fatty acids). In addition, vegetable oils may contain some contaminants: pesticides, trace metals, mineral oil aromatic hydrocarbons (MOAH), aflatoxins, dioxins, polycyclic aromatic hydrocarbons (PAH) and traces of organic solvents. However, one of the main disadvantages of refining is the loss of substances responsible for the healthy and technological properties of the oils, such as tocopherols, phospholipids, squalene, polyphenols and phytosterols [52]. The two main industrial processes for vegetable oil refining are chemical and physical refining, as described in Figure 11.

The difference between these two processes lies in the method used to remove the free fatty acids. In chemical refining, the free fatty acids are removed by adding caustic soda and separating the soap by centrifugation (mechanical separation), while in physical refining, the free fatty acids and other compounds are removed in the final step by distillation under a high vacuum with steam injection [51].

#### 4.2.2. Batch Extractors

Early industrial plants used batch extractors, which remained the only devices for solvent oil extraction for several decades. Batch extractors are simple systems with a seed inlet and a meal outlet. The extraction occurs through maceration, with the extraction solvent being pumped into the extractor, which is usually equipped with an agitator. At the end of the extraction, the miscella is drawn off and the meal is desolventised in the extraction vessel by indirect steam heating, sometimes combined with a live steam injection. The resulting vapours leave the vessel through the ventilation system and are fed into the condenser. Batch extraction is no longer used in large-scale plants, while it continues to be used for small production such as the extraction of specialty oils. Some disadvantages of batch extractors can be overcome if several batch extractors are combined into one group extractor (Figure 12) [8].

In this way, a semi-continuous process is achieved that also allows countercurrent extraction. With this configuration, each time a batch of miscella is discharged from one extractor, it is pumped to another extractor containing material previously extracted with an oil-richer miscella. Stage I contains the raw seed and Stage II contains extracted seed, whereas Stage III contains the furthest extracted seed. The extraction solvent is pumped from III to I countercurrent. The same principle of successive extraction cycles in continuous countercurrent extractors is obtained. An additional vessel for feeding fresh seeds allows for a semi-continuous system (Figure 12c).

#### 4.2.3. Countercurrent Continuous Extractors

In the immersion process, the seed is completely immersed in the solvent. The static system requires stirring to ensure the exchange of the locally concentrated solvent. Immersion extractors are particularly suitable for the extraction of fibre-rich oilseeds. An example of a countercurrent immersion extractor is the Crown Model IV (Figure 10). This extractor model is designed for the immersion extraction of powdery materials that cannot be extracted in percolation extractors. In this device, the material itself moves through the successive solvent tanks by means of an inclined conveyor that slowly draws the still immersed material into the solvent pool. The gentle movement and rotation of the bed minimises the formation of fines. The solids rise above the solvent level as the inclined conveyor rotates around the top pulley, and then fall freely through the solvent into the second pool. The densities of the material and the solvent must be sufficiently different to allow the particles to fall. This process is repeated for all the extraction stages. Above the last inclined ramp, fresh solvent is added, and the mixture is fully aspirated at the point where the solids enter the extractor. The last inclined ramp allows the solvent retained by the material to drain off before it leaves the extractor [50].

The percolation process is based on the principle of permanent wetting of the surface by percolating solvent. During percolation, there is a constant exchange between the free-flowing solvent and the solvent trapped or absorbed by the material. This phenomenon ensures that locally oil-rich solvent is permanently replaced by fresh or low-concentration solvent. The process requires an effective pre-treatment of the seed to obtain as many open cells as possible. Compared to immersion, the material does not have to be stirred up, thus avoiding a further undesirable reduction in size. In addition, only a limited portion of the fine particles mix with the miscella, as the material acts as a filter and retains them. In optimised countercurrent percolation systems, a miscella oil content of 30% can be achieved [8]. There are different types of percolation extractors: rotary, perforated belt, sliding-bed and rectangular loop extractors.

In rotary extractors there are usually five stages of extraction, followed by a sixth one where fresh hexane is used for the final wash. Ideally, there should be a slight preponderance of liquid over solids in each stage. This ensures that all solids are in contact with the liquid. If the solids bed allows solvent to flow too quickly so that no liquid column can be maintained, the solvent/mixture must be evenly distributed over the material. The preparation of the material must allow adequate percolation flow. A too slow flow may result in flooding of the cell with the mixture overflowing into the next one. The presence of channels in the bed of an inhomogeneous material can divert the solvent flow via the path of least resistance. Usually, in a deep bed extractor, flooding is more likely than too rapid percolation. Extrusion of the material can be useful as it agglomerates the fines and creates a more uniform bed of material [5]. To achieve good contact between the matrix and solvent, some models of deep bed extractors are equipped with a chamber in which the material is immersed in the solvent before extraction. This process reduces the risk of channelling during the percolation, where the solvent flows along preferential channels and does not adequately contact certain areas of the bed. Shallow bed units are usually loaded dry. The most common rotary extractor supplied today is the Reflex Extractor (Desmet Ballestra Group, Paris, France), where the material is mixed with the miscella and fed into the rotating baskets. This design constantly reduces the transport of solvents in the spent material to the desolventiser, saving steam energy [53].

In a perforated belt extractor, the material bed is a continuous mass extending the length of the extractor. In these systems, percolation occurs through an endless loop perforated belt that keeps the bed of material moving. The belt transports the incoming material through the extractor stages, each consisting of a sump below the belt and miscella spray heads above. The residence time is adjusted depending on the belt speed. The belt is slightly inclined. This helps the miscella to flow countercurrent through the material more efficiently. The incoming material, mixed with concentrated miscella, is deposited on the belt to form a bed of material. The depth of the bed can vary but is usually in the middle range (1.5–2.0 m deep). As with the other extractor types, there are several extraction stages and a final wash with fresh hexane. Each miscella pump returns the miscella to the same stage from which it came. The material moves countercurrent to the miscella that overflows from one sump chamber to the previous one. The most common perforated belt extractor supplied today is the LM™ Extractor (Desmet Ballestra Group, Paris, France).

A sliding-bed extractor pushes the solids along a fixed steel plate with a special groove that allows the miscella to pass through while the materials are retained. The depth of the material bed can be adjusted during operation and is usually between 0.5 and 1.3 m. The extractor consists of two overlapping plates. The moving cell assembly wraps around a drive pulley, causing the oilseed contents to be turned over and pushed onto a second perforated plate, then making a second pass through the extraction chamber. Countercurrent miscella washes are introduced onto the bed of cells as they pass under miscella spray heads [12]. Sliding-bed extractors process flaked material, extruded collets or prepressed cake [5]. The most common sliding-bed extractor supplied today is the Lurgi Extractor (JJ-Lurgi Engineering, Selangor Darul Ehsan, Malaysia).

In rectangular loop extractors, the material is drawn through a closed ring-shaped chamber. The housing closes around itself in a spiral. The solid material is deposited in a shallow bed, usually less than 1.0 m, and travels a distance equivalent to 50 times the depth of the bed [5]. Fresh oilseeds are conveyed through an inlet to the upper level and sprayed with a rich mixture. The oilseeds undergo a multi-stage (usually seven) countercurrent extraction as they pass through the ring. External valves allow each pump to return part of the miscella to the stage from which it came and part to the previous stage. Sometimes the entire miscella is recycled to the stage from which it came to extend the contact time and maintain a sufficient liquid height above the material. The final wash occurs using fresh solvent, followed by a dripping period of up to about 30% solvent by weight before the extracted material comes out [12]. The most common rectangular loop extractor supplied today is the Crown Type III Extractor (Crown Iron Works, Minneapolis, MN, USA).

#### 4.2.4. Conventional Solvent Extraction

Hexane is a petrochemical solvent used for the extraction of vegetable oils and also to produce flavours and fragrances, natural extracts, pharmaceuticals and food supplements. The global demand for hexane in these sectors is about 1.1 Mt per year, of which 650 Kt are used for oilseed extraction and 450 Kt to produce natural extracts and specialty oils [47]. In recent years, a shift in consumer awareness has been observed worldwide, reflecting growing concern about the health and sustainability of production systems. The same trend is also reflected in recent policies, in particular in the European Union (EU), with the adoption of the European Green Deal in 2020, which aims at climate neutrality, a circular economy and a toxic-free environment. In this context, the new Safe and Sustainable-By-Design (SSbD) standard aims to ensure that chemicals, materials and products are designed, manufactured and used in a way that does not harm humans and the environment [54].

Among the solvents exploited at industrial level for the extraction of non-polar edible natural products such as dyes, flavours, fragrances or lipids, hexane is undoubtedly the most used, and it appears on the list of extraction solvents allowed for foods or food ingredients production in the EU (Directive 2009/32/EC) [46]. This substance has been known for more than fifty years for its neurotoxicity and reproductive toxicity, but new evidence has shown that hexane is also a potential endocrine disruptor. *n*-hexane (the main isomer of hexane) is classified as STOT RE 2 under the Regulation on Registration, Evaluation, Authorisation and Restriction of Chemicals (REACH), which means that it is suspected of causing organ damage through prolonged or repeated exposure. Currently, a process is underway to reclassify *n*-hexane from STOT RE 2 to STOT RE 1, i.e., from suspected to proven to be neurotoxic to humans.

In our recent work, we presented applications of hexane in food products and ingredients’ extraction and evaluated new evidence of its toxicity. Finally, some alternatives to hexane for the extraction of natural products were listed [47]. At present, research and development of environmentally friendly and safe extraction methods is essential for the gradual replacement of toxic solvents such as hexane. Green solvents are supposed to be non-toxic or low-toxic, safe to use and handle, effective and derived from renewable resources. They generally meet some of the 12 principles of Green Chemistry established by Anastas and Eghbali [55] and the 6 principles of Green Extraction established by Chemat et al. [56].

Directive 2009/32/EC contains a first list of solvents for which no conditions are specified, and a second list for which permitted uses and maximum residue limits are set. Among the solvents listed in Directive 2009/32/EC, five of the seven solvents on the first list are compatible with oil extraction and are therefore potential alternatives to hexane. These include butane, ethyl acetate, ethanol, carbon dioxide and acetone [13]. Of these five solvents, ethanol has the most extensive research literature. The main alternative solvents and methods to conventional oil extraction with hexane are presented in Table 3.

## 5. Green Solvents’ Extraction

### 5.1. Alcohols

Ethanol has been most studied in the literature as a possible alternative to hexane for oil extraction. Not only is it safe for humans, but it can also be obtained from biological resources without producing toxic waste. Among the solvents listed in Directive 2009/32/EC, ethanol has a high acceptance and is the only solvent except for water and CO_2_ that is included in the list of solvents allowed in organic animal feed production.

The main problems related to oil extraction with ethanol are the low oil solubility (especially at low temperatures and in the presence of water) and the high latent heat of vaporisation (846 kJ/kg compared to 333 kJ/kg for hexane), which requires more energy for distillation. The low solubility of the oil in ethanol can be exploited to separate the oil after extraction by simply cooling the solvent. In this way, the oil can be recovered without the need to evaporate the entire solvent, thus reducing the costs associated with the higher energy required to regenerate the solvent [13]. Johnson and Lusas [57] proposed that the energy requirement of the entire process can be reduced by about 25% compared to the corresponding process with hexane, although it is not clear whether the drying of the oilseeds was taken into account. The problem with cold separation is the formation of a third emulsified phase containing gums at the interface between the solvent and oil phases. Hron and Koltun [58] proposed a method to improve the decanting of miscella from ethanol extraction. The heterogeneous solution is processed through a phase separator where free and emulsified oil and gum are separated from oil-lean miscella. The proposed method involves heating of the interface to 38 °C, a temperature at which the gums lose some of their entrapped mixture, agglomerate and sink to the bottom of the oil phase. Then, the oil and gum phases are treated with caustic soda and centrifuged to produce semi-refined oil [58]. Oliveira et al. [59] investigated the composition of the two oil- and alcohol-rich phases formed during the cooling of the miscella as it leaves the extractor. The authors predicted an ethanol recovery of up to 98% when the mixture is cooled from 80 °C to 25 °C.

Another technique for solvent separation involves the use of ethanol-compatible membranes. The miscella can be significantly concentrated using nanofiltration membranes [60]. The concentrated miscella could significantly reduce the energy required for distillation. Theoretically, then, it can be assumed that the energy cost for ethanol distillation is not an obstacle, as this step can be bypassed.

The composition of the extract obtained with ethanol differs from that of the oil obtained with hexane. The extraction of polar lipids (e.g., phospholipids) is more effective with ethanol due to its polar nature [49]. Similarly, more polar lipids can be extracted when isopropanol is used, while more triacylglycerols are extracted when hexane is used. The solubility of oils in ethanol has been studied for most commercial oils; Figure 13 shows the solubility of oil in absolute ethanol and isopropanol and azeotropic mixtures as well as of sunflower oil in various ethanol/water mixtures.

At the azeotropic concentration (95.6%), ethanol is saturated with about 1% oil at 40 °C and 8% at 80 °C, slightly above its boiling point at atmospheric pressure. In addition, the presence of free fatty acids has a significant influence on the solubility of oil in alcohols, so that these saturation values may vary. In general, at the temperatures achievable in industrial extraction plants, the oil concentration in the miscella is unlikely to exceed 8–10%. More solvent is therefore needed to extract the same amount of oil as with hexane (S/L ratio of 3–4 versus about 0.8–1 with hexane), resulting in higher process costs.

The most recent studies on the extraction of the major vegetable oils with alcohols are listed in Table 4. Sawada et al. [62] showed that an increase in water content in the ethanol strongly suppressed soybean oil extraction, while an increase in temperature favoured it. The opposite behaviour was observed for proteins: an increase in the water content of the solvent enhanced the extraction of these compounds, while increasing the temperature decreased the protein content in the meal. The fatty acid profile of the ethanol-extracted oils showed a composition typical of soybean oil, regardless of the extraction conditions. Toda et al. [63] described the extraction kinetics of soybean oil and free fatty acids using ethanol with different degrees of hydration (0 and 5.98% mass of water) at temperatures of 40, 50 and 60 °C. Increasing the degree of ethanol hydration suppresses the extraction of soybean oil but increases that of FFA, while temperature favours the solubility of both fatty compounds. Bessa et al. [64] successfully used a multi-batch solid–liquid extraction system to simulate continuous countercurrent ethanol extraction of rice bran oil. The theoretical number of equilibrium steps required for the extraction of rice bran oil was higher for ethanol extraction than for hexane extraction. Ethanol extraction required working with a higher solvent to solid mass ratio. However, ethanol can extract the solid matrix completely in five steps (residual oil less than 0.5% in the meal). The fatty acid composition of the crude oil shows that the oil extracted with ethanol has the typical composition of rice bran oil. Capellini et al. [49] showed how the water content in the solvent and the temperature of the process strongly affected the properties of the protein fraction of the meal. The nitrogen solubility index decreased from about 40% in anhydrous solvents to 17% and 15% in aqueous ethanol and isopropanol, respectively. Minor nutraceutical compounds (e.g., γ-oryzanol, tocopherols and tocotrienols) were recovered more efficiently in oil extracted with aqueous ethanol and isopropanol than with hexane.

However, several obstacles make the ethanol non-competitive with hexane. The first is the need to dry the oilseeds (water content below 2–3%) to avoid water entering the system, which is dissolved by the ethanol, reducing its affinity to the oil. Conventional drying is associated with high energy costs. Secondly, the affinity of the matrix for ethanol is much higher than for hexane, which means that a higher quantity of solvent is retained and must be evaporated from the meal. Ethanol extraction also results in the denaturation of proteins, which lose their solubility on prolonged contact with this solvent. The displacement of structural water results in a conformational change due to the merging of domains with opposite electrostatic charges and the formation of new bonds [13]. However, this loss of solubility is reversible after some hydration treatments, unlike that occurring when the proteins are exposed to heat. Thirdly, there is a significant risk of protein degradation in the meal due to the mass of solvent to be evaporated and the higher boiling point of ethanol. In addition, the use of direct steam should be avoided in order not to rectify the solvent at each extraction cycle. Desolventisation technologies other than those used in the existing plants should be used. These include the use of larger heating surfaces to work only with indirect steam or vacuum desolventiser. Both solutions are associated with higher capital investments. Fourth, the affinity of the solvent to the matrix, the difficulty of using direct steam and the need to maintain temperatures compatible with protein stability result in a much higher solvent content in the meal than in the case of hexane. This can lead to increased flammability of the meals.

Carré [13] proposed a process in which the dehulled seeds are first cold pressed to mechanically extract a high-quality oil and then extracted with azeotropic ethanol to complete the defatting. Subsequently, a washing step with aqueous ethanol removes soluble carbohydrates, phenolic compounds and most glucosinolates. The meal is pressed to mechanically remove as much of the retained solvent before a gentle desolventisation. This model is more expensive than the current one, but the different products obtained (pressed oil, solvent-extracted oil, meal and an ethanol extract of lecithin, phenolic compounds, etc.) could make it economically attractive. Carré et al. [69] proposed a cost estimation to evaluate the feasibility of a large-scale hot ethanol extraction process. The simulated extractor involved extraction at a temperature above the boiling point (90–95 °C) in a module consisting of hydrocyclones operating in partial countercurrent. The total cost of processing was estimated at EUR 47.4 per tonne of seed processed, which was higher than the cost of a conventional hexane process (EUR 30/tonne). The additional profit margin after processing costs (EUR 14.64 more than the hexane process for 1 tonne of seed) is mainly due to a better valuation of the meal and oil and a lower phospholipid and pigment content. In addition, in contrast to the conventional process with hexane, extraction with hot ethanol produces molasses, whose value was estimated at 70% of the meal price. However, the greatest uncertainty of the simulation relied on the performance of the process under real countercurrent extraction and solvent regeneration conditions.

Potrich et al. [70] simulated the replacement of hexane by ethanol (hydro or anhydrous) as a solvent for the extraction of soybean oil. Different solvent recovery methods (simple distillation and extractive distillation with glycerol or monoethylene glycol) were evaluated for ethanol. An economic analysis and life cycle assessment of the process were carried out. The economic analysis showed that hexane possessed a net present value (NPV) about 10.2% higher than that of the best-case ethanol process (hydrous). However, replacing hexane with ethanol resulted in a lower global warming potential (GWP) by avoiding the emission of about 10,600 tonnes of CO_2_ equivalent per year in an industry that crushes 125 tonnes of soybean per hour.

Citeau et al. [68] evaluated the extraction of rapeseed oil with aqueous ethanol (92% and 96% by weight, respectively) and isopropanol (84% and 88% by weight, respectively) and compared them with hexane as a reference. The alcoholic solvents, together with the oil, extracted 11–15% non-lipid substances, increased the protein concentration to 42–43% and reduced the glucosinolates concentration to 7–19% in the meal, depending on the type of alcohol. In comparison, protein and glucosinolates concentrations after extraction with hexane were 38% and 25% in the meal, respectively. Alcohol extraction increased the protein content of the meal by 13% compared to hexane extraction but greatly reduced the solubility of the protein. The increased water content improved the extractability of the glucosinolates. Isopropanol with the highest water content reduced the glucosinolates concentration by 49–73% compared to meal extracted with other alcohols.

Isopropanol is another alcohol used in the extraction of vegetable oils. It is generally obtained by chemical synthesis, while production by fermentation is limited. However, isopropanol is more expensive than other alcoholic solvents, such as ethanol. Its main advantage is that oils are more soluble in this solvent than in ethanol, even in the presence of water (Figure 13), and it would be possible to obtain miscella with 20% oil concentration. Compared to ethanol, isopropanol has a higher boiling point of 4 °C, but its latent heat of vaporisation of 666 kJ/kg is only 78% of that of ethanol, and its dielectric constant is lower (18.6 vs. 26.5) [13]. On the other hand, the azeotropic water contents are different with 12.3% (isopropanol) and 4.4% (ethanol) by mass. Compared to ethanol, this solvent can represent a relatively easy substitute for hexane as it remains effective in the presence of water. However, it incurs additional costs compared to the hexane process. Furthermore, the literature lacks data on the effects of this solvent on rapeseed and sunflower seeds.

Comerlatto et al. [71] suggested that for ethanol-rich mixtures, mass transfer is limited by convection at the solid–liquid interface, while for isopropanol-rich mixtures, internal diffusion limits the extraction process. An economic analysis considering solvent costs at different extraction yields evidenced that isopropanol is more suitable for low-yield extractions (less than 70%) and ethanol for high-yield extractions.

According to the work by Li et al. [65], the extraction of oil from coarsely ground rapeseed was more promising using isopropanol (83.1%) and butanol (78.3%) than ethanol (22.8%) (analytical grade solvents). The oil extracted in isopropanol was 94.7% TAG. Perrier et al. [67] obtained a higher extraction yield with isopropanol than with aqueous ethanol (96%), even with the use of ultrasound. Compared to the diffusivity of oil in hexane, the diffusivity in isopropanol and ethanol was slower. In another study by Sicaire et al. [66], the extraction of rapeseed with ethanol showed a similar yield to that obtained with isopropanol (technical grade solvents). However, the extract in isopropanol contained large amounts of polar lipids (only 80.19% of TAG). Citeau et al. [68] reported a lower rapeseed oil extraction yield with isopropanol at 87.8% (89.3% of yield), compared to hexane (93.5%) and ethanol at 95.6% (92.7%). A small increase in water content significantly reduced the extraction yield of alcohol solvents. Due to the variability in extraction conditions (extraction technique, temperature, liquid/solid ratio, agitation, etc.), sampling conditions (moisture content, drying efficiency, oilseeds’ pre-treatment) and oilseed composition, it is difficult to compare the different authors’ experimental work.

### 5.2. 2-Methyloxolane

2-Methyltetrahydrofuran, also known as 2-methyloxolane (2-MeOx), is a biodegradable solvent derived from levulinic acid or furfural obtained from lignocellulosic biomass conversion (e.g., corncobs and sugarcane bagasse). 2-MeOx shows interesting properties that are technically comparable to those of hexane and could easily be transported on an industrial scale [72]. 2-MeOx is predominantly lipophilic (log *p* = 1.85) and can therefore dissolve both fatty molecules, such as hexane (log *p* = 4.00), and more polar molecules, due to the presence of an oxygen atom (dipole moment = 1.38 D). Its boiling point (80 °C) is high enough to allow a good extraction temperature but low enough to be easily removed from the final products and recycled. In addition, its density (0.855) and viscosity (0.6 cP) are close to those of hexane and within an acceptable range for efficient diffusion through solid particles [73]. Moreover, 2-MeOx has a much safer toxicological profile than hexane, and the use of 2-MeOx in industrial production could lead to a 97% reduction in CO_2_ emissions compared to petroleum-based solvents [72].

2-MeOx has already been approved for the extraction of organic and natural cosmetic ingredients (COSMOS label) and pharmaceutical products. On 20 March 2022, EFSA declared 2-MeOx a safe solvent for food applications, based on a comprehensive review of scientific studies. Methyloxolane (EcoXtract^®^) was added to the list of permitted solvents for food and feed production in Europe on 26 January 2023 (Directive 2009/32/EC) [74,75]. The main studies on the extraction of vegetable oils with 2-MeOx are listed in Table 5.

The partial miscibility of 2-MeOx with water can lead to better diffusion in cases where the solids to be extracted contain moisture. 2-MeOx forms an azeotrope with water (10.6% water and 89.4% 2-MeOx) and a second distillation is required to obtain the dry solvent. In the recovery of 2-MeOx in industrial extraction plants, the recovered organic phase is saturated with water after liquid/liquid separation. The 2-MeOx/H_2_O mixture (95.5/4.5% at 55 °C) must be fed to another distillation step to recover the dry 2-MeOx. However, studies have shown that 2-MeOx 95.5% can be used directly for oil extraction, showing a similar efficiency to the dry solvent [77,78]. Using COSMO-RS and Hansen solubility predictions, Sicaire et al. [66] showed that 2-MeOx can be considered the best alternative to *n*-hexane among all solvents tested, as it can efficiently dissolve the desired compounds. The extraction yield with 2-MeOx was comparable to that with hexane on a laboratory scale (45.96 and 46.34 g/100 g DM, respectively).

Using the 6-litre pilot percolation extractor, the percentage of residual oil in the rapeseed meal was 1.8% for hexane and 0.8% for 2-MeOx [76]. In addition, the extraction with 2-MeOx was faster, as three washing cycles allowed the extraction of almost 95% of the total oil, while five washes were required to extract 96% of the total oil for hexane. The energy evaluation of the industrial extraction process showed that the total amount of energy was slightly higher with 2-MeOx (365 MJ/t seed) than with hexane (284 MJ/t seed) [76]. Sicaire et al. [66] obtained a rapeseed oil yield of 47.19% with 2-MeOx and 46.7% with hexane, but the extract with 2-MeOx had a lower percentage of TAGs (92.83% and 99.12%, respectively) due to the extraction of more polar lipids [66]. Claux et al. [77] investigated the use of dry 2-MeOx and 2-MeOx 95.5% as alternatives for the extraction of soybean oil. Due to their higher polarity, dry 2-MeOx (23.5%) and 2-MeOx 95.5% (23.7%) gave higher oil yields than hexane (18.8%). The yield differences were ascribed to the co-extraction of more polar additional compounds such as phospholipids and isoflavones. However, the protein dispersibility index (PDI) and KOH protein solubility were slightly lower for meal extracted with dry 2-MeOx and 2-MeOx 95.5%. The concentrations of antinutritional factors in the meals were the same after extraction with hexane and 2-MeOx.

Annual production of 2-MeOx is estimated at about 4500 tonnes, which is far below the amount needed for its continuous large-scale use. 2-MeOx trades at EUR 8.0/kg for purchases over 100 tonnes from Pennakem (Memphis, Tennessee), and hexane at EUR 0.9/kg [79]. A price reduction would increase the competitiveness of this solvent. The second economic disadvantage is the higher energy consumption for evaporation of the solvent from the miscella and desolventisation of the meal compared to extraction with hexane. However, the yield with 2-MeOx is higher, and with further optimisation of recycling and the reduction in losses, the estimated total cost of the process increases by only EUR 0.47 per tonne of extracted seed. This difference is minimal and could easily be compensated by a slight increase in the retail price of 2-MeOx-extracted oils: one cent per kilo, or 1 per cent of the premium, could actually bring more profit than hexane [72].

Among the various solvents here presented, 2-MeOx, ethanol and SFE-CO_2_ are the ones at the most advanced stage of study. Rapinel et al. [72] reported industrial-scale studies that demonstrated the scalability of the process. The extraction of 46 tonnes of canola press cake (22.8% oil) at 340 kg/h in an industrial immersion extractor allowed a very good extraction performance (0.3% residual oil). The replacement of hexane by 2-MeOx does not require any significant plant changes. The only adjustments encompass the replacement of incompatible polymeric materials, the replacement of the wastewater boiler with a distillation column to maximise the recovery of the solvent present in the aqueous phase and the installation of a thermostatically controlled settling tank to reduce the amount of 2-MeOx in the aqueous phase. An advantage is also the possible conversion of the mineral oil absorption system replaced by cold water.

Further studies on the extraction of secondary metabolites, antinutrients and protein quality with 2-MeOx are needed. The refining process of crude oils extracted with 2-MeOx also requires further research.

### 5.3. Supercritical and Subcritical Fluids

Supercritical fluids are an alternative to conventional organic solvents. A fluid reaches a critical state when it is simultaneously heated and pressurised above its critical pressure. The main advantages of supercritical fluid extraction (SFE) include the absence of explosion hazards, higher selectivity in the extraction of neutral lipids, shorter extraction times, easy solvent recovery that preserves the oil and meal from thermal and oxidative degradation, and the avoidance of halogenated organic solvents. On the other hand, due to the high initial cost of equipment installation, and the energy and capital costs associated with regular maintenance, SFE is more commonly used in industrial processes that aim to obtain a product with a high economic value, resulting in a high-quality lipid extract.

Carbon dioxide (CO_2_) is the most widely used solvent for SFE, being inert, non-toxic, non-flammable, inexpensive, abundant and endowed with moderate critical properties (Tc = 31.1 °C, Pc = 7.38 MPa). CO_2_ is a generally recognised safe solvent (GRAS), so products containing extracts made from food-grade CO_2_ are safe for human health. The use of supercritical CO_2_ extraction techniques allows the efficient extraction of lipophilic compounds at generally mild temperatures (30–70 °C), without leaving traces in the extract. Since supercritical CO_2_ is a non-polar solvent, its solubilising capacity is known to be between that of pentane and toluene [47]. The solubility of oil in CO_2_ is strongly dependent on pressure and temperature. At about 300 bar, it is between 0.3 and 0.7%, with temperature having a negative effect (raising the temperature from 40 to 80 °C leads to lower oil solubility) [80]. Only above 350 bar is there a positive relationship between solubility and temperature. As a result, in today’s systems (which are generally limited to 3–400 bar), a large amount of solvent must be used to extract the oil. Oil separation from the solvent should also be carried out under supercritical conditions to save energy. The solubility of triglycerides in CO_2_ is so low under 160 bar that excellent separation is possible. Another advantage of this system is that there is no need for a condenser to liquefy the CO_2_ ahead of the pump. Cavitation in the plunger pump due to the phase change of the CO_2_ is also avoided. Specific mass flows between 10 and 50 kg CO_2_/kg·h at surface velocities of the solvent between 1 and 5 mm/sec have been used for SFE of oilseeds [81].

SFE plants comprise four main components: (1) a pump to ensure the volumetric flow of the fluid; optionally, it can be preceded by a cooler to transport the gaseous components in a liquid state, (2) a heat exchanger, (3) an extractor, where static and dynamic extractions take place by modulating the pressure regulated by a valve, and (4) a separator [82].

To improve the solubility of polar compounds, a polar co-solvent such as methanol or ethanol can be added. Moreover, systems have been developed that integrate ultrasonic treatments sequentially or simultaneously with SFE. Duarte et al. [83] described a system with an ultrasound probe internally coupled into the supercritical extraction cell. To avoid the use of CO_2_ at high pressures, binary gas mixtures were investigated. A mixture of CO_2_ and propane allows complete miscibility of triglycerides below 300 bar and at temperatures up to 70 °C [81]. Ultra-high-pressure SFE (pressures greater than 70 MPa) severely limits the amount of solvent to be used and the extraction time. Recent studies have shown that extracts obtained under high pressure are enriched with important ingredients that can only be obtained under these conditions. Although the investment costs are higher, the operating costs are lower at higher pressure. This is due to the shorter extraction times resulting from the higher oil solubility and the extraction of high-value compounds. Depending on the application, ultra-high-pressure SFE could be competitive [79].

The use of liquid CO_2_ has emerged as an innovative extraction technique that offers many of the same benefits as supercritical CO_2_, but at lower pressures (≤15 MPa) and temperatures (about 25 °C), therefore reducing energy costs. In addition, due to its low polarity compared to most organic solvents, liquid CO_2_ might have higher selectivity for neutral lipids, while affinity for non-neutral lipids is limited. The overall extraction yield is usually lower than that of organic solvents due to the lower polarity. The use of co-solvents such as methanol is also compatible with this technique [84].

Liquified gas extraction (LGE) is a promising alternative for the extraction of lipids and other compounds. LGE involves extraction with gaseous organic solvents under pressure, which allows the solvent to be in a condensed state. Unlike SCF, LGE requires moderate pressures (1–10 bar), which results in lower energy consumption and facilitates industrial-scale application. Depending on the pressure and temperature conditions, the dissolving power can be adjusted by changing the solvent density, which enables selective extractions. The most common liquid gases are propane, butane and dimethyl ether (the first two are listed in the Directive 2009/32/EC). The main advantages of LGE are the following: (1) they can be used at low temperatures: due to their low boiling point, liquefied gases can be vaporised at moderate temperatures, preserving oil and meal and leaving no traces of solvent residues; (2) they can be used at moderate pressures; (3) under “normal” solid–liquid extraction conditions, most liquefied gases are chemically inert [47].

In LGE, the temperature increase has a positive effect on the extraction yield up to a certain limit (e.g., 55 °C at 0.5 MPa for *n*-butane). Above these temperatures, the solubility of the oil decreases as the gasification rate of the solvent increases [2]. The main studies on the extraction of vegetable oils with supercritical and subcritical fluids are listed in Table 6. Some studies have compared SFE and LGE in the extraction of oilseeds. For example, Pederssetti et al. [85] analysed the extraction of canola seeds using SFE-CO_2_ and LGE-propane. For SFE, increasing pressure had a positive effect, while increasing temperature had a negative effect on the extraction yield. For LGE extractions, the effect of temperature was more pronounced compared to pressure due to the low variation in density with changing pressure over the experimental range examined. LGE with propane shows faster extraction kinetics than SFE since propane solubilises triacylglycerols better than CO_2_. The oxidative stability and fatty acid profile of the extracted oil were similar for the two solvents. However, the oil yields obtained with SFE and LGE were only 52.7% and 64.3% of those obtained with hexane Soxhlet extraction. Due to the lower oil yield, the protein content of the meal was also lower with SFE (29.9%) and LGE (29.9%) than with hexane extraction (36.7%). Sun et al. [86] compared canola meal extracted with SFE-CO_2_ with pressed cake and meal extracted with hexane. Both the hexane-extracted and SFE-CO_2_-extracted meals had a higher protein content than the pressed cake. The glucosinolates content was lower in the meal extracted with SFE-CO_2_, while the phosphorus content was higher than in the meal extracted with hexane. The phenolic acid contents of the meal extracted with hexane and SFE-CO_2_ were similar. The addition of ethanol as a co-solvent during SFE-CO_2_ reduced the phosphorus and phenolic acid content in the meal. It is well known that the addition of ethanol enhances phospholipid extraction with SFE-CO_2_. Both meals extracted with hexane and SFE-CO_2_ showed a similar protein solubility, high water-holding capacity, high oil absorption and high emulsifying capacity. Boutin and Badens [87] investigated the influence of different parameters on the oil extraction yield for rapeseed and sunflower seeds using SFE-CO_2_. The authors found that pressure and extraction duration are the most influencing parameters together with temperature in the case of rapeseed. Under optimal conditions, SFE-CO_2_ yielded 89.9% and 78.5% of oil from rapeseed and sunflower, respectively. SFE-CO_2_ proved to be a selective process as no traces of phospholipids were detected in the extracted oil. This would make it possible to avoid degumming during oil refining. Nimet et al. [88] showed that LGE-propane afforded higher yields of sunflower oil than SFE-CO_2_, working at a lower pressure and with a shorter extraction time. The fatty acid profile of the sunflower oil and the protein content of the meal were not significantly affected by the solvents or operating conditions. The values for the tocopherol content and oxidative stability of the oil obtained during extraction with LGE-propane and SFE-CO_2_ were higher than those obtained with conventional extraction with hexane. In another study by Rapinel et al. [89], LGE of sunflower oil with *n*-butane at low pressure (0.2 to 0.4 MPa) resulted in a lower extraction yield compared to hexane extraction (36.9% vs. 53.4%).

In general, oils extracted with LGE have a higher quality (lower acid value), a higher content of phytosterols and carotenoids (antioxidants), a better shelf life (lower iodine value) and a better oxidation stability than oils extracted with conventional techniques. The fatty acid profiles of the oil extracted with LGE or SFE do not differ from those of the conventional methods [2]. The separation of oil from the miscella after LGE can also be performed by membrane filtration. This technique can lead to energy savings (reduced use of steam, full or partial replacement of traditional degumming, refining and bleaching steps). An ideal membrane for solvent recovery must combine specific properties such as high oil retention and permeate fluxes suitable for industrial scales, as well as heat resistance and mechanical and chemical compatibility with the process [93].

Currently, the largest SFE-CO_2_ plant is in Korea and is dedicated to the extraction of sesame oil for food purposes (capacity of 2 × 2600 L/550 bar). A techno-economic analysis of the SFE-CO_2_ performance for the extraction of pequi pulp oil showed that the production costs in pilot (100 L) and industrial (500 L) plants were attractive in eight scenarios tested. The most promising one showed a production cost of about EUR 33/kg pequi oil, with the purchase of pequi and other feedstocks accounting for about 85% of the cost. Moreover, the payback period was estimated as less than one year, which is attractive as the initial investment can be recovered quickly [79].

Fiori [94] developed a model to predict the economic feasibility of an industrial scale SFE-CO_2_ plant (three extractors in series operating in countercurrent) designed to extract 3000 tonnes of grape seeds per year. The authors suggested that switching from a single mode to a cascade of two extractors increases the extraction efficiency from 83% to 86%. The break-even point that makes the process economically viable is a value of 5.9 EUR/kg for grape seed oil extracted with SFE-CO_2_. According to these results, SFE-CO_2_ proves to be a cost-effective alternative for the extraction of high-value-added specialty oils.

However, the high pressures required to reach the supercritical state limit its use to high-value-added products, as the investment costs for an industrial plant are correspondingly high. A clear disadvantage of SFE-CO_2_ is also the need to extract large quantities of material in a semi-continuous process by filling and emptying the extraction vessels at atmospheric pressure. In conclusion, the use of SFE-CO_2_ in the extraction of lipids for the large-scale production of biofuels and commodity vegetable oils is an uneconomic alternative, as the high costs of installation and maintenance, combined with energy costs, would significantly increase the price of the final product [79].

For liquefied gases’ pilot-scale extraction, the company CELSIUS Sarl (Villette de Vienne, France) developed an isobaric LGE process (NECTACEL extractor) where the system always stays at liquid/vapor equilibrium. The liquefied gas is evaporated in the boiler at the same pressure (isobaric mode) and the vapours rise naturally to the condenser for solvent regeneration. No solvent pump or compressor is required to run the cycle, including solvent injection or recycling. The absence of mechanical equipment means lower energy consumption and maintenance costs. The extraction and separation steps are carried out at room temperature, which benefits heat-sensitive or thermo-oxidisable molecules. Solvation properties can be improved by using a co-solvent with complementary ionic character. The most used solvents are butane, fluorocarbons (HFO1234ze) and dimethyl ether. Extraction and separation take place at a relatively low pressure (less than 10 bar), with no technological capacity limit for the equipment. CELSIUS built several prototypes with 1 L, 200 L and 500 L capacities. The main applications of this system concern the extraction of molecules for pharmaceutical, cosmetic (flavours and perfumes) and nutraceutical purposes [95].

### 5.4. Terpenes

Terpenes (also called isoprenoids or terpenoids) are acyclic, bicyclic or monocyclic hydrocarbons formed biosynthetically from isoprene units (C_5_H_8_). Terpenes are a large and extensive family of natural products (over 30,000) with relatively different physical properties. They are generally considered biodegradable solvents with low environmental impact and low toxicity, and they are mainly extracted from tree leaves and agricultural sources. In addition, microbial production of terpenes has enormous potential as it avoids dependence on natural resources. For example, limonene, which is generally obtained from the by-products of citrus juice production by steam distillation and condensation, can be produced by microbial bioconversion of glucose by *E. coli* or *S. cerevisiae* [82]. *D*-limonene, α-pinene and *p*-limonene have been investigated as alternative solvents for the extraction of vegetable oils.

Terpenes have molecular weights and structures suitable to substitute for *n*-hexane. The solubility parameters of the solvents were investigated using Hansen’s parameters and COSMO-RS (Conductor-like Screening Model for Realistic Solvents) simulations. They gave similar results for terpenes and *n*-hexane for dissolving TAGs as well as sterols and tocopherols. However, when their dielectric constants are considered, terpenes are slightly more polar and show a higher dissociation force than *n*-hexane. In terms of safety, they have a higher flash point and are therefore less flammable and dangerous. Although the results in the laboratory scale are encouraging, the scalability of the terpene extraction process is limited. The main disadvantage of using terpenes is their high viscosity and density, and the higher energy consumption in solvent recovery by evaporation due to their higher boiling points (155–176 °C) and higher enthalpies of evaporation (37–39 kJ/mol) compared to *n*-hexane (Bp = 69 °C, ΔHvap = 29.74 kJ/mol) [66,76,96].

Heteroazeotropic distillation of the binary terpene–water mixture (about 50% *v*/*v*) leads to evaporation of the terpene and separation of the terpene–water distillation product by phase decantation. With this system, the solvent can be recovered at a lower temperature (about 97–98 °C at atmospheric pressure), and a high recovery rate of terpenes can be achieved (about 90% with 98% purity for α-pinene) [97]. The main studies on the extraction of vegetable oils with terpenes are listed in Table 7.

Bertouche et al. [97] showed a higher yield with α-pinene compared to hexane when extracting soybean oil (21.1 vs. 19.1 g/100 g DM) and sunflower oil (67.2 vs. 52.6 g/100 g DM). The fatty acids extracted from both solvents were equivalent in terms of identified compounds and relative proportions. The higher boiling point of α-pinene reduces the viscosity of the oil, which leads to better diffusion through the matrix. Li et al. [65] evaluated the use of different terpenes (*p*-cymene, d-limonene and α-pinene) to extract rapeseed oil. The three solvents afforded higher oil extraction yields than *n*-hexane (88.9%, 80.8% and 65.5%, respectively, compared to 58.2% for *n*-hexane). The composition of the oil extracted with *p*-cymene contained more FFA and DAG than *n*-hexane-extracted oil, and the tocopherol and tocotrienol content was lower. Sicaire et al. [66] reported lower yields with *p*-cymene (39.71%) and d-limonene (36.94%) compared to hexane (46.71%) for rapeseed oil extraction. The percentage of TAG in the oil extracted with *p*-cymene (82.03%) and d-limonene (51.31%) was significantly lower than with hexane (99.12%). *D*-limonene and other unsaturated terpenes are readily oxidised in air and the resulting oxidation products are labelled as allergens. *p*-menthane (stable saturated derivative of d-limonene) and cis/trans-pinane: 7:3 (stable saturated derivative of α-pinene and β-pinene) were investigated as new saturated terpene solvents for oil extraction. Both solvents showed promising results with comparable oil yields to *n*-hexane (40.5 vs. 39.5 g/100 g DM for *p*-menthane and *n*-hexane, respectively [99], and 42.5 vs. 43.2 g/100 g DM for pinane and *n*-hexane, respectively [98]). The encouraging lab-scale results still need further research for the development of a pilot-scale terpene extraction process.

### 5.5. Alternative Hydrocarbon Solvents

Among the alternative hydrocarbon solvents, isohexane, cyclohexane and *n*-heptane have been proposed to replace hexane. These solvents also originate from the petrochemical industry, and although they are considered less harmful to health than hexane, comprehensive studies are lacking, especially on their chronic toxicity [100,101]. In contrast to *n*-hexane, the C6 isomers without *n*-hexane have not shown neurotoxic effects in animal experiments. This is attributed to the fact that no neurotoxic 1,4-diketones are formed during the metabolism of these hexane isomers. However, further studies on the chronic effects of exposure to these substances are needed (e.g., lack of information on teratogenicity and genotoxicity) [102].

Wan et al. (1995) compared five hydrocarbon solvents (heptane, isohexane, neohexane, cyclohexane and cyclopentane) with hexane for the extraction of cottonseed oil. Extraction with the six hydrocarbon solvents at their respective boiling points showed that normal paraffins (hexane and heptane) were more efficient and significantly better than cyclohexane or branched paraffins in extracting oil from cottonseed flakes. Cyclohexane extracted 95.4% of the oil, with respect to normal paraffins, while cyclopentane, isohexane and neohexane extracted 91.6%, 88.8% and 88.8%, respectively. The laboratory-scale extraction study (S/L ratio of 1 to 5.5 *w*/*w*, temperature 10–45 °C below the boiling point of the solvent) showed that hexane removed 100% of the oil from the flakes at 55 °C after a one-step extraction; heptane extracted 100% at 75 °C and 95.9% at 55 °C; and isohexane extracted 93.1% at 45 °C. The extraction efficiencies of cyclopentane, cyclohexane and neohexane were lower. Based on these results, heptane and isohexane are the most suitable alternative hydrocarbon solvents to replace hexane [103].

These two solvents were therefore investigated as potential substitutes for hexane in a cottonseed processing plant with a capacity of 300 tonnes/day. The extraction efficiencies of isohexane and heptane, as measured by extraction time and residual oil in the meal, colour of refined and bleached oil and solvent loss, were comparable to that of hexane. However, isohexane appears the best solution to replace hexane with minimal equipment adjustment. The higher boiling point of heptane increased the energy consumption of the desolventiser/toaster (D/T) system, reducing the daily production by 20–30%. Heptane would require more D/T capacity to achieve the same tonne production as hexane. In addition, the solvent loss of heptane was 12% higher than the average annual loss of hexane. However, due to the high boiling point of heptane, it can be used in a wide temperature range from ambient to 80 °C.

For isohexane, the D/T processing rate was 10–20% higher than for hexane. Isohexane boils at 55 °C and therefore has a fairly narrow operating temperature range. This lower extraction temperature may also affect the extraction efficiency of isohexane, as shown by the slightly higher residual oil content in the meal compared to hexane. However, with isohexane, daily production increased by more than 20% and gas consumption decreased by more than 40% [104]. Isohexane is more expensive than hexane because of the additional isomerisation process to produce it. The improved production and lower energy requirements for isohexane should offset the solvent cost difference. Isohexane was shown to be a viable choice to replace hexane in oilseed extraction, but the current need for solvents that are safe and not derived from petroleum makes this solvent out of step with the new frontiers defined by green extraction.

### 5.6. Other Organic Solvents

Acetone, ethyl acetate, cyclopentyl methyl ether (CPME) and dimethyl carbonate (DMC) have also been tested as alternative solvents to hexane for the extraction of lipids from oilseeds. Of these solvents, only acetone and ethyl acetate are included in Directive 2009/32/EC.

Acetone has been used mainly for the extraction of cotton oil, but its use in the extraction of oilseeds is limited. Acetone ranks between hexane and ethanol in terms of selectivity towards lipids. This solvent is completely miscible with water and its volatility is higher than that of hexane due to its lower boiling point, which is not an advantage from a safety point of view. Acetone has a relatively low latent heat of vaporisation (534 kJ/kg) and does not form an azeotrope with water, which allows easy recovery by distillation. As with ethanol and isopropyl alcohol, the oil extracted with acetone can be recovered by simply cooling the miscella. On the other hand, phospholipids are not soluble in acetone, which facilitates refining but does not allow recovery of the lecithin. The presence of water in acetone significantly reduces its ability to extract the oil. In addition, acetone can reduce the amount of the gossypol and aflatoxins in the meal. One of the main problems with acetone extraction is that it usually imparts a very unpleasant odour (cat’s urine smell) to the meal. This smell is due to the reaction of mesityl oxide (which is obtained from acetone by condensation reactions) with hydrogen sulphide (coming from the breakdown of one or more of the sulphur-containing amino acids). Further drying of the flakes and solvent before extraction can reduce these unpleasant odours but significantly increases production costs [105].

Ethyl acetate is a colourless ester of acetic acid and ethanol, characterised by low toxicity and a pleasant odour. Ethyl acetate is one of the solvents approved for food production (Directive 2009/32/EU) and is safer and cheaper (33% less) than hexane. It has a similar boiling temperature and latent heat of vaporisation, but the main disadvantages of using ethyl acetate are its higher viscosity, density and dielectric point than hexane [106]. This solvent is partially miscible with water (8.5% at 20 °C) and forms with it an azeotrope with a boiling point lower than that of the pure compound (70 °C). Ethyl acetate seems to have been relatively little studied compared to other solvents. The researchers who have made comparisons seem to prefer isopropanol. However, as the economic environment has changed, the impact of energy costs may tip the balance in favour of a solvent that is easier to distil. Lohani et al. (2015) used an accelerated pressure extractor (at 1500 psi) to extract seed oils (rapeseed, flaxseed) using ethyl acetate and hexane for comparison. Ethyl acetate shows a lower viscosity and polarity under subcritical conditions, affording an oil yield (120 °C, 90 min) close to that of Soxhlet extraction with hexane, reaching 40.38% for rapeseed and 33.33% for flaxseed, compared to 42.96% and 35.62% with hexane, respectively. These results evidenced that the quality (heating value, density, viscosity, fatty acid profile and unsaponifiable matter) of the oils extracted from different oilseeds with ethyl acetate is almost equivalent to that of the oils extracted with hexane [106]. The use of solvent mixtures such as water/ethanol/ethyl acetate was investigated for the extraction of rapeseed oil, obtaining the maximum rate of oil yield (32.35 mg/mm) with the mixture containing 9.17%, 6.61% and 84.17% of water, ethanol and ethyl acetate, respectively [107].

Dimethyl carbonate (DMC) is a non-polar aprotic solvent with good miscibility with water (139 g/L), and it is readily biodegradable in the atmosphere and non-toxic. Currently, the most common commercial route to produce DMC is the oxidative carbonylation of methanol using O_2_; however, new alternative processes have been developed to produce DMC from CO_2_. DMC showed a good extraction performance for triglycerides and could also be used as a mild solvent for biocatalyst-assisted biodiesel conversion [108].

Although cyclopentyl methyl ether (CPME) is currently synthesized by petrochemical-based routes (using cyclopentene and methanol to obtain CPME with excellent atom economy), some bio-based alternatives paved the way for a future biogenic source of this solvent. In particular, several biomass-based processes focused on the production of chemical precursors that can be used for CPME production, such as cyclopentanone or cyclopentanol [109]. CPME is endowed with interesting properties, such as low solubility in water (1.1 g CPME/100 g), the formation of a positive azeotrope and low enthalpy of vaporisation, as well as stability in strongly acidic or basic conditions. CPME also shows low toxicity, a negligible peroxide formation rate and a narrow explosion range. However, it has a high flash point (compared to hexane) and a higher boiling temperature (106 °C). The main studies on the extraction of vegetable oils with other organic solvents are listed in Table 8.

Sicaire et al. [66] investigated the use of ethyl acetate, CPME and DMC as alternatives to hexane for the extraction of rapeseed oil by conventional Soxhlet extraction. All these three solvents gave similar oil yields (42.83%, 42.77% and 41.53% for ethyl acetate, DMC and CPME, respectively), but these values were about 10% lower than those of hexane (46.71%). Furthermore, HPTLC analysis showed that the oils extracted with these solvents had different compositions. Only 88% of the extracted components were triglycerides (TAGs), compared to 99% of the oil extracted with hexane.

### 5.7. Enzymatic and Surfactant-Assisted Aqueous Extraction

The aqueous extraction of oil from oilseeds involves the diffusion of lipids in water and the subsequent formation of emulsions. The emulsified oil can be separated by demulsification, e.g., through temperature change or using enzymes. Aqueous oil extraction is an environmentally friendly and safe process that allows the simultaneous production of edible oil and protein isolates or concentrates in the same process. This process also brings economic advantages as it avoids the use of organic solvents, effectively removes antinutritive factors and toxins from the meal, and avoids the need for solvent evaporation and for degumming of crude oils [110]. The main limitations are the lower efficiency of the aqueous oil extraction, the necessary demulsification of the extract to recover the oil, the wastewater treatment, and the lowest protein content and quality of the meal. To improve the yield of the aqueous processes, enzymes have been used to facilitate the oil release. These mainly hydrolyse the structural polysaccharides (cellulose, hemicellulose and pectin) that form the cell wall, or the proteins of the cytoplasm or of the lipid bodies’ membrane.

Selected enzymes were tested on different types of oilseeds and achieved much higher extraction yields than the original aqueous process (in some cases over 90%) [111]. Several factors influence the efficiency of aqueous enzyme extraction. Pre-treatment to reduce the size of oily materials by grinding or flaking increases the extraction yield due to there being better accessibility for the enzymes. The type of enzyme is a key factor in the whole process. The use of enzymes alone or in combination can be determined based on the structure of the oilseed, the types of enzymes and the experimental conditions. Due to the high protein content of soybean, the use of proteases led to more efficient oil extraction. In the extraction of rapeseed, whose cell wall consists mainly of pectin, treatment with pectinase resulted in an 85.9% increase in oil yield. The use of enzyme mixtures in some cases showed better performance than single enzymes, probably due to a synergistic effect. Since for each enzyme there is a certain optimal pH value, special attention must be paid to optimising this parameter. In general, higher extraction efficiency is obtained at pH values far from the isoelectric point. For example, low oil yields were achieved in soybeans, rapeseed and sunflower seed extractions at the isoelectric point due to low protein solubility.

Temperature also plays an important role as enzyme activities are generally effective at 45 °C or less and an increase in temperature can lead to the denaturation of proteins, reducing the release of oil from the oilseed. In general, increasing the concentration of the enzyme (up to about 1% enzyme by weight of raw material) until the active sites of the substrate are saturated results in greater degradation of the desired product and greater oil recovery. Further increases in enzyme concentration are not only not cost-effective, but can also induce off flavours, bitterness and caramelisation of sugars that can interfere with the oil extraction process. Adequate agitation allows partial breakdown of the mechanical barriers (cell wall) and uniform mixing of the entire contents of the reaction mixture. However, too intensive stirring is not only energetically costly but can also form stable emulsions that are difficult to separate. In general, oils obtained by aqueous enzyme extraction contained low levels of phosphatides and peroxides and similar levels of free fatty acids compared to oils extracted by conventional methods [112].

The optimisation of the entire process must not only focus on achieving the highest possible oil yield but must also consider the ease of splitting the resulting emulsion. In particular, some parameters such as oil droplet size and viscosity of the bulk emulsion are very useful in evaluating emulsion stability [111]. In addition, some proteins show an emulsifying capacity that generally increases during enzymatic proteolysis until a certain degree of hydrolysis is reached, and then decreases. Therefore, the extent of the enzymatic proteolytic action must be optimised to obtain both a higher oil yield and a less stable oil in water emulsion. The main studies on the extraction of vegetable oils with aqueous enzymatic extraction are listed in Table 9.

Cheng et al. [121] assessed the environmental impacts of soybean oil production through an extruding–expelling process, hexane extraction and aqueous extraction. Hexane extraction showed the highest environmental impact due to the use of an organic solvent. Extrusion led to the highest greenhouse gas and criteria pollutant emissions due to the high energy required for heat-pressing processes. Finally, enzyme-assisted aqueous oil extraction had similar environmental impacts as the extrusion process but also reduced greenhouse gas and criteria pollutant emissions.

Enzyme-assisted oil extraction was also studied in combination with other green technologies such as microwaves and ultrasound. Enzyme-assisted extraction combined with microwaves, in addition to the general extractive benefits of microwaves, accelerates the enzyme reactions and delays the denaturalisation of the enzymes compared to conventional heating. Similarly, the use of ultrasound in combination with enzymatic extraction can increase oil yield and improve its quality and stability over time [4].

The necessary drying of the wet meal to be stored and the limited recycling of the enzymes represent additional costs in this process [4]. Immobilisation of the enzyme on a solid support is a possible strategy that allows many cycles of oil extraction and reduces costs [110]. Aqueous enzymatic extraction has good potential for simultaneous oil and protein recovery from oilseeds. However, the application of this technology is still hampered by the high cost for the production of enzymes, the lower extraction yields compared to solvent extraction, the long incubation time and the additional de-emulsification process.

Surfactant aqueous oil extraction (SAOE) is another aqueous extraction technique that seems to be a promising method on a laboratory scale and allows an oil yield of slightly more than 90%. The basic concept of SAEP is to use surfactants to reduce the interfacial tension (IFT) between oil and water (about 19–24 mN/m), making aqueous oil extraction possible [122,123]. Surfactants reduce the IFT due to their amphiphilic structure, which gives them the property to adsorb at the oil–water interface. Compared to conventional surfactants, extended surfactants contain polar intermediate groups, such as polypropylene oxide and/or polyethylene oxide, inserted between the hydrophilic head and the lipophilic tail. This molecular structure gives surfactants the ability to achieve a very low IFT (≤10^−2^ mN/m). Lowering the IFT to such low values allows the formation of so-called microemulsions, which are ideal systems for the co-solubilisation of oil and water. In addition, salts (e.g., NaCl, CaCl_2_ and KCl) and alcohols (from ethanol to medium-chain alcohols such as octanol and decanol) are often added to surfactant solutions to improve their interfacial properties and thus reduce both the critical micelle concentration and the surface or interfacial tension. SAOE involves two main steps: extraction and separation. In the first step, the oil is extracted by immersing the matrix in the surfactant solution, usually with stirring. The oil and water form a turbid mixture that must be separated and purified to obtain a free oil phase. First, the solid phase is separated from the liquid phase by centrifugation. Then, in the liquid phase of interest, three layers are generally observed: (i) a layer of free oil, (ii) a microemulsion of oil in aqueous solution and (iii) the surfactant solution. In order to recover the oil from this liquid fraction, the system must be destabilised, whereby the oil–water–surfactant system is relatively stable. Gagnon et al. [122] described in detail the properties of extended surfactants, the state of the art of SAOE and the limitations that need to be overcome for the industrial application of this extraction method. The authors concluded that although initial results have shown this technology to be effective on a laboratory scale, it is still far from industrial application. From a technical point of view, the fractionation process of the oil–water–surfactant mixture, the recycling of the surfactants, the removal of salts and dehydration of the residual cake after extraction are the main critical points. In addition, the energy costs of the process should be taken into account and more information on the quality of the oil and solid fractions should be collected and compared with those obtained in the conventional way. Finally, the identification of new extended-like, bio-based and environmentally friendly surfactants would be desirable.

### 5.8. Extraction Optimization Methods

Oil extraction represents one of the most critical steps in oilseed processing, as it determines the quality and quantity of the oil obtained. By carefully optimising the extraction conditions for each extraction method, yield and quality can be improved. In addition, a proper optimisation process can save time and energy, thus reducing the cost of the entire oil extraction process. While conventional extraction methods are highly optimised today, especially at the industrial scale, new extraction technologies require extensive optimisation. Nde and Fonchaet [1] gave an overview of techniques for optimising the extraction of oil from plant material.

Most of these methods involve the use of statistical software to optimise control variables (e.g., extraction time, solids/solvent ratio, type of solvent, seed pre-treatment, etc.) to reduce costs and increase performance while avoiding degradation of product quality as much as possible. Two main methods for process optimisation can be identified in the literature: the single parameter optimisation method and the response surface methodology (RSM).

In single parameter optimisation, the best conditions for oil extraction are determined for a single factor at a time, holding the other factors constant. In this case, the ideal level of a factor that gives the best response is defined. However, this laborious procedure does not allow the interactions between factors to be considered. The RSM adopts several mathematical and statistical techniques to jointly model and analyse the effects of two or more process factors that have an impact on the desired response. A major step in optimization is screening the factors studied to have the significant effects of the analytical method. RSM reduces the number of experimental trials but maintains the expected accuracy and determines the responses for the interactive effect of multiple factors. However, RSM is limited by fitting the data to a second order polynomial, but not all systems with curvatures are compatible with this model. RSM is based on the relationship between product properties and regression equations that describe the correlations between input parameters and product properties. In many extraction procedures, the regression coefficient R^2^ and/or the absolute/standard error of deviation are used to validate the models used. A model is considered good if R^2^ > 0.70 and/or error < 10%. Among the many RSM software available, Design experts, Sigma Plot version 12.5, MATLAB version 5.0, IBM SPSS and Minitab version 15.5 have been commonly used to optimise oilseed extraction [1].

## 6. Conclusions and Perspectives

In the oilseed industry, the individual operations of storage, crushing and extraction are considered interdependent, and the impact of each step on the overall process must be evaluated. From this point of view, the introduction of new technologies can significantly improve the outcome of the different stages, and therefore the quality of the final product. Since extraction remains the most critical step because of its impact on oil quality and quantity, new methods of gas- and solvent-assisted mechanical extraction have been proposed. The wide availability and low cost of petroleum-based solvents made them the best candidate for any industrial application. However, their harmful effects on the environment and health have recently stimulated research towards more environmentally friendly, non-toxic, bio-based solvents. The technology readiness level (TRL) of green solvents varies widely; while some have already been studied at semi-industrial or even industrial scale (SFE-CO_2_, ethanol and 2-MeOx), only laboratory-scale results are available for others (aqueous enzymatic extraction, DMC, ethyl acetate, CPME). Further studies are therefore necessary to assess the feasibility of using them on a larger scale. In addition, to meet the huge demand from the industry, highly efficient and cheaper production methods of green solvents need to be developed. The properties of the green extracts (composition, stability, etc.) should also be further investigated to ensure their similarity or improvement compared to conventional products obtained with petroleum-based solvents. Finally, features and advantages of bio-based solvents should be included in educational programs (students, extraction industry and government agencies) to raise awareness and promote the transition to a greener economy. The industrial use of bio-based solvents will indeed reduce human health risks and protect the environment, promoting the valorisation of agricultural waste, according to a circular economy approach.

## Figures and Tables

**Figure 1 molecules-28-05973-f001:**
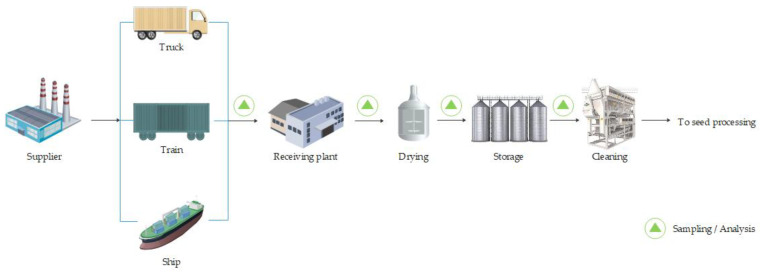
Typical processes in the reception and storage of oilseeds.

**Figure 2 molecules-28-05973-f002:**
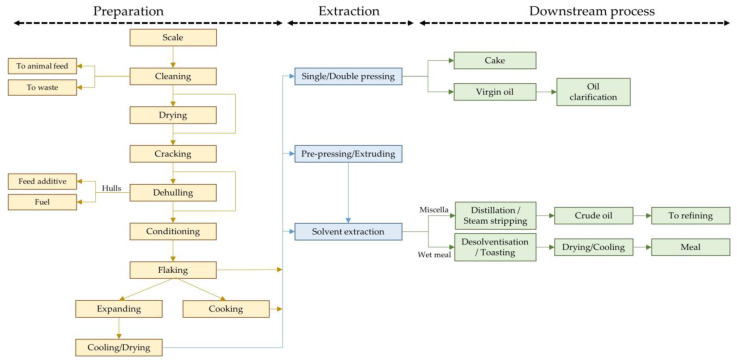
Oilseed general processing.

**Figure 3 molecules-28-05973-f003:**
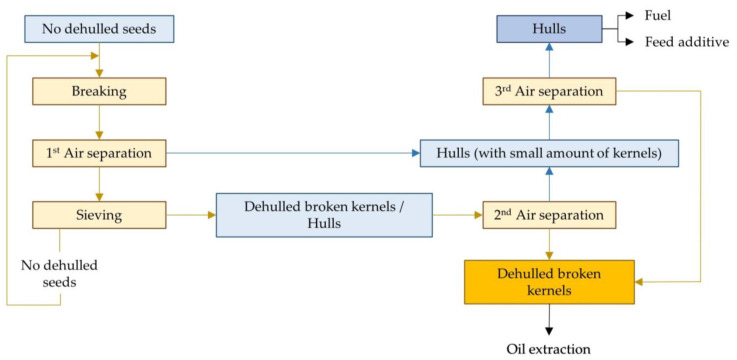
Soybean general dehulling process.

**Figure 4 molecules-28-05973-f004:**
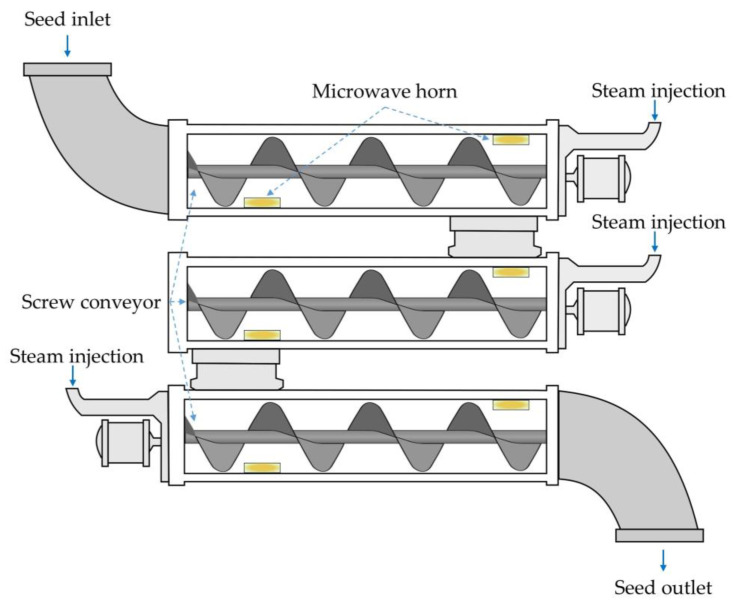
Example of a continuous industrial-scale plant for microwave pre-treatment of oilseeds.

**Figure 5 molecules-28-05973-f005:**
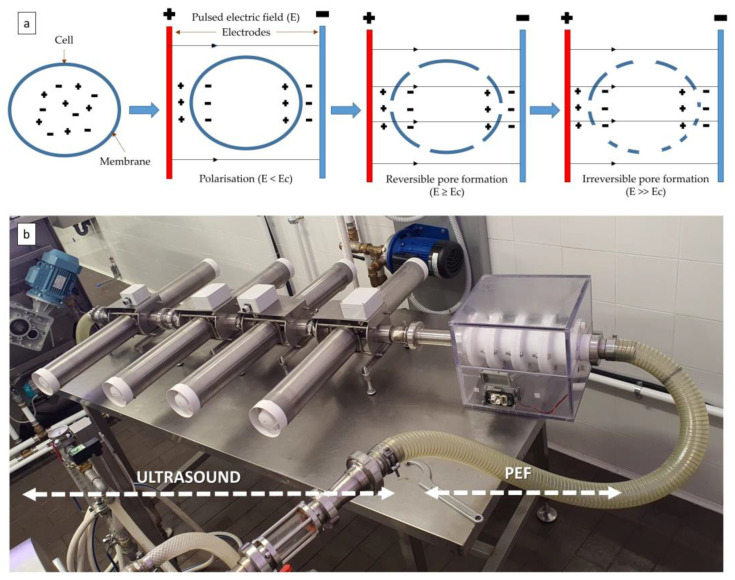
(**a**) PEF electroporation mechanism; (**b**) continuous flow pre-treatment and/or extraction system under combined US (C2FUT Srl) and PEF (Energy Pulse Systems) conditions, DSTF-University of Turin [25].

**Figure 6 molecules-28-05973-f006:**
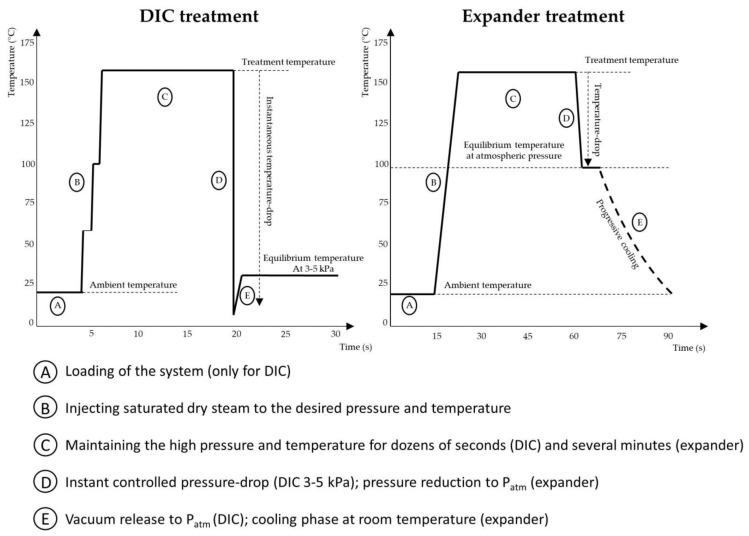
Comparison between DIC and expander treatments [17].

**Figure 7 molecules-28-05973-f007:**
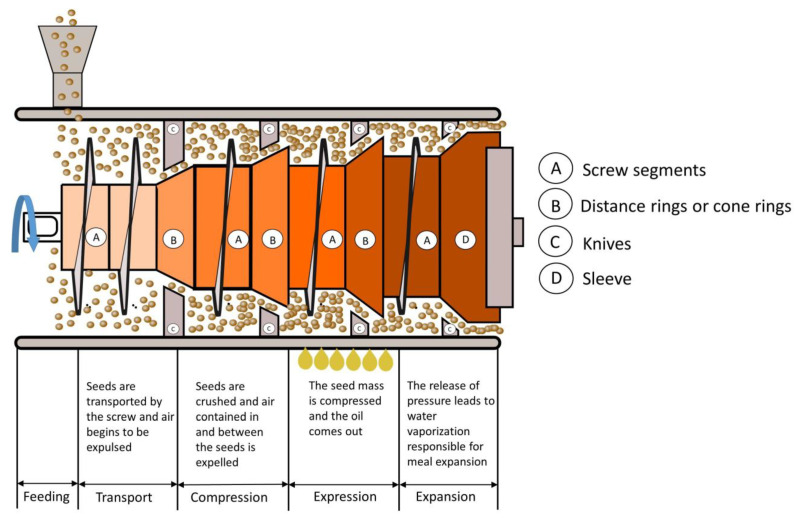
Screw expression process of oilseeds.

**Figure 8 molecules-28-05973-f008:**
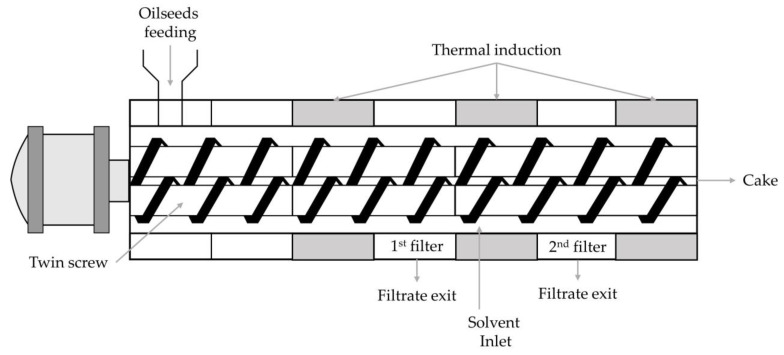
Schematic representation of the twin-screw extruder combining thermomechanical extrusion and solvent extraction [42].

**Figure 9 molecules-28-05973-f009:**
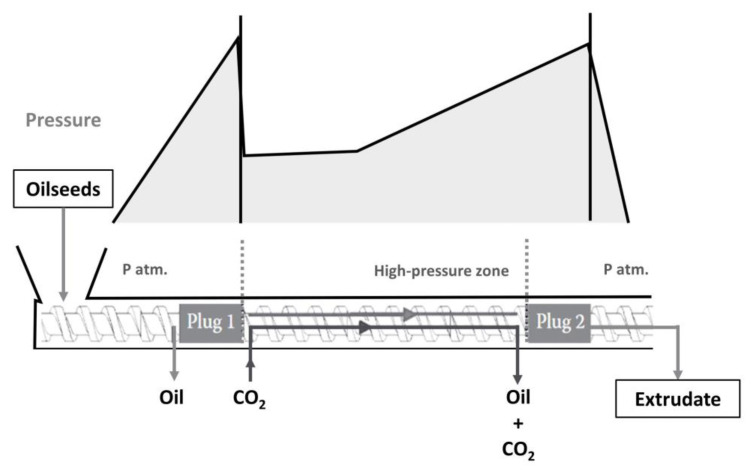
Possible two-stage extruder design for GAME [43].

**Figure 10 molecules-28-05973-f010:**
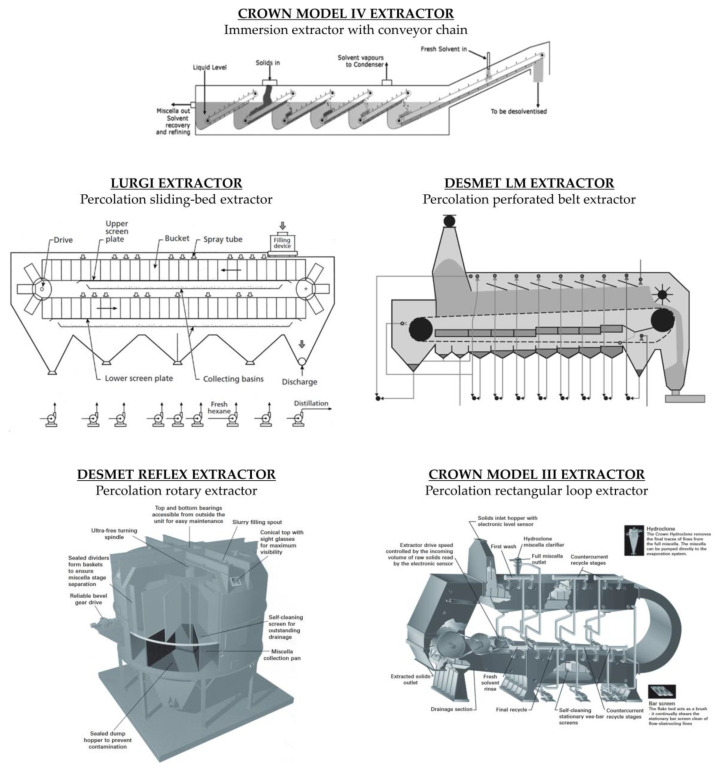
Countercurrent immersion and percolation extractors [3,12,39,50].

**Figure 11 molecules-28-05973-f011:**
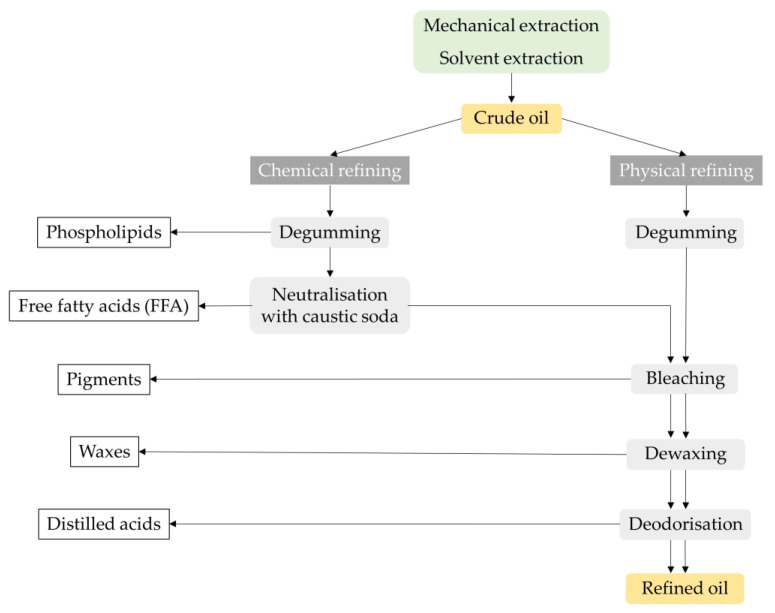
Crude oil chemical and physical refining process: a general overview.

**Figure 12 molecules-28-05973-f012:**
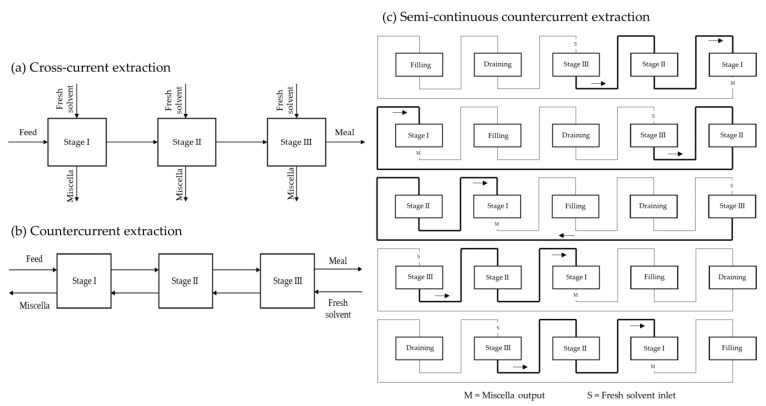
(**a**) Cross-current extraction principle; (**b**) countercurrent extraction principle; (**c**) semi-continuous countercurrent extraction.

**Figure 13 molecules-28-05973-f013:**
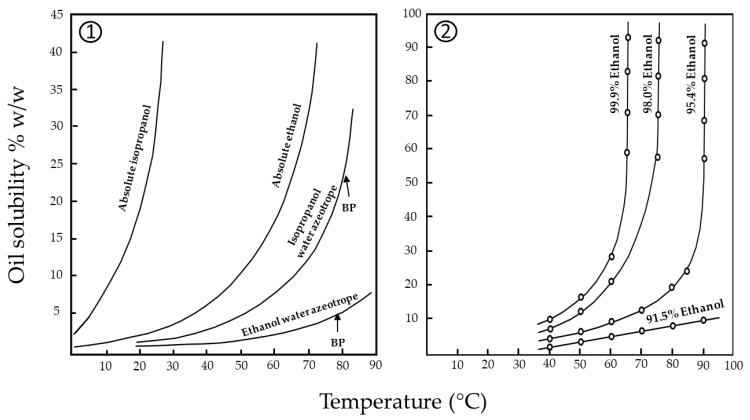
(**1**) Oil solubility in absolute ethanol and isopropanol and azeotropic mixtures; (**2**) sunflower oil solubility in ethanol/water mixtures [61].

**Table 1 molecules-28-05973-t001:** Vegetable oil production in 2018/2019 (adapted from Mwaurah et al. [4]).

Oil Type	Year Production (Mt)	Portion of World Production (%)	Main Producing Country
Palm	69.6	35.0	Indonesia
Soybean	57.1	28.8	China
Rapeseed	27.8	14.1	China
Sunflower	17.8	9.0	Ukraine
Palm kernel	8.0	4.0	Indonesia
Peanut	5.5	2.8	China
Cottonseed	5.2	2.6	China
Coconut	3.4	1.7	Philippines
Olive (virgin)	3.1	1.6	Spain
Total	198.3	100.0	

**Table 2 molecules-28-05973-t002:** Influence of process parameters and seed pre-treatment on pressing performance [38].

Parameters	Barrel Temperature	Maximum Barrel Pressure	Capacity	Cake Residual Oil
*Press parameters*				
↓ Choke opening	↑	↑	↓	↓
↓ Screw rotation speed	↓	↑	↓	↓
*Seed pre-treatment*				
↑ Heating	↑	↑	↑	↓
Flaking	↓	↑	↑	↑
Flaking + heating	↑	↑	↑	↓
↑ Seed moisture content	↓	↓	↓	↑

**Table 3 molecules-28-05973-t003:** Relevant physicochemical properties of various solvents.

Properties	*n*-Hexane	Ethanol	Isopropanol	*n*-Butanol	2-MeOx	Acetone	CPME	Ethyl Acetate	DMC	*D*-Limonene	α-Pinene	*p*-Cymene
Molecular formula	C_16_H_14_	C_2_H_6_O	C_3_H_5_O	C_4_H_10_O	C_5_H_10_O	C_3_H_6_O	C_6_H_12_O	C_4_H_5_O_2_	C_3_H_6_O_3_	C_10_H_16_	C_10_H_16_	C_10_H_14_
Molecular weight (g/mol)	86.18	46.07	60.10	74.12	86.13	58.09	100.16	88.11	90.08	136.23	136.24	134.22
Dipole moment (D)	0.08	1.69	1.66	1.75	1.38	2.91	1.27	1.88	0.18	2.44	2.58	2.34
Density (g/mL)	0.65	0.79	0.79	0.81	0.85	0.79	0.86	0.90	1.07	0.83	0.88	0.86
Boiling point (°C)	69	78.4	82.5	117.8	80.2	56.53	106	77.1	90.3	176	156	174
Azeotrope boiling point (°C)	61	78.1	80.4	92.4	71	-	83	70.4	-	-	-	-
Melting point (°C)	−95	−114	−89	−89	−136	−95	−140	−4	4.6	−74	−62	−68
Solubility in water (g/100 g at 20 °C)	immiscible	miscible	miscible	9.1	14	miscible	1.1	8.5	immiscible	1.38	0.25	2.34
Partition coefficient (Log *p*)	4.00	−0.18	0.16	0.88	1.85	−0.24	1.59	0.73	0.23	4.20	4.83	4.10
Enthalpy of vaporization ΔH (kJ/kg)	333	846	666	582	373	534	289	430	414	326	262	284
LD50 Oral (rat mg/kg)	25,000	7060–9000	4710–5840	790–4360	>2000	5800–9800	200–2000	10,200	13,000	4750	3700	3

2-MeOx, 2-Methyloxolane; CPME, cyclopentyl methyl ether; DMC, dimethyl carbonate.

**Table 4 molecules-28-05973-t004:** Vegetable oils’ extraction with alcohols.

Material	Seeds’ Pre-Treatment	Extraction Technique	Solvent	Oil Extraction (% of Total Oil)	Ref.
Rapeseed	Coarsely ground	Soxhlet extraction	Ethanol	22.8	[65]
			Isopropanol	83.1	
			Butanol	78.3	
Soybean	Cracked, dehulled, flaked and expanded	Batch extraction 60 °C	Ethanol absolute	90.3–93.4	[62]
		Batch extraction 90 °C	Ethanol absolute	89.0–92.5	
		Batch extraction 60 °C	Ethanol 94%	28.2–31.3	
		Batch extraction 90 °C	Ethanol 94%	92.5–96.0	
		Batch extraction 60 °C	Ethanol 88%	2.9–6.6	
Rapeseed	Coarsely ground	Soxhlet extraction	Ethanol	46.6 ^a^	[66]
			Isopropanol	45.0 ^a^	
Rapeseed	Flaked	Batch extraction	Isopropanol 99%	63	[67]
		Batch extraction with US	Isopropanol 99%	79	
		Batch extraction with US	Ethanol 96%	51	
Rapeseed	Crushed	Four stage cross-current extraction	Ethanol 96%	92.7	[68]
			Ethanol 92%	86.5	
			Isopropanol 88%	89.3	
			Isopropanol 84%	87.1	

^a^ Oil yield (g/100 g DM seed).

**Table 5 molecules-28-05973-t005:** Vegetable oils’ extraction with 2-MeOx.

Material	Preparation	Extraction Technique	Oil Yield (g/100 g DM Seed)	Ref.
Rapeseed	Coarsely ground	Soxhlet extraction	47.2	[66]
Rapeseed	Finely ground	Soxhlet extraction	46.0	[76]
	Rapeseed cake	Pilot Soxhlet extraction	95.6 ^a^	
Soybean	Finely ground	Soxhlet extraction	23.5	[77]

^a^ Oil extraction (% of total oil).

**Table 6 molecules-28-05973-t006:** Vegetable oils’ extraction with supercritical and subcritical fluids.

Material	Preparation	Extraction Technique	Solvent	Oil Yield (g/100 g DM Seed)	Ref.
Flaxseed	Ground	Laboratory-scale extractor	SFE-CO_2_ 99.9%, 30 MPa, 50 °C	35.3 ^a^	[90]
Sunflower	Milled	Laboratory-scale extractor	Propane 99.5%, 8 MPa, 60 °C	92.7 ^b^	[88]
			Propane 99.5%, 12 MPa, 60 °C	100 ^b^	
			Propane 99.5%, 10 MPa, 45 °C	90.2 ^b^	
			SFE-CO_2_ 99.9%, 25 MPa, 40 °C	100 ^b^	
			SFE-CO_2_ 99.9%, 22 MPa, 50 °C	75.6 ^b^	
			SFE-CO_2_ 99.9%, 25 MPa, 60 °C	87.8 ^b^	
Rapeseed	Milled	Laboratory-scale extractor	Propane 99.5%, 8 MPa, 60 °C	62.5 ^b^, 23.1	[85]
			Propane 99.5%, 12 MPa, 60 °C	64.4 ^b^, 23.8	
			Propane 99.5%, 10 MPa, 45 °C	55.9 ^b^, 20.7	
			SFE-CO_2_ 99.9%, 25 MPa, 40 °C	52.7 ^b^, 19.5	
			SFE-CO_2_ 99.9%, 22 MPa, 50 °C	48.1 ^b^, 17.8	
			SFE-CO_2_ 99.9%, 25 MPa, 60 °C	49.2 ^b^, 18.2	
Flaxseed	Ground	Laboratory-scale extractor	*n*-Propane 99.5%, 8 MPa, 60 °C	28.6	[91]
			*n*-Propane 99.5%, 12 MPa, 60 °C	28.8	
			*n*-Propane 99.5%, 10 MPa, 45 °C	28.2	
Sunflower	Ground	Pilot-scale extractor	*n*-Butane 95%, 0.2 MPa, 20 °C	36.6 ^a^	[89]
			*n*-Butane 95%, 0.28 MPa, 30 °C	36.7 ^a^	
			*n*-Butane 95%, 0.37 MPa, 40 °C	36.9 ^a^	
Flaxseed	Ground	Laboratory-scale extractor	*n*-Butane, 99.5%, 0.5 MPa, 54 °C	28.8 ^a^	[92]

^a^ Oil yield (g/100 g initial material); ^b^ oil extraction (% of total oil).

**Table 7 molecules-28-05973-t007:** Vegetable oils’ extraction with terpenes.

Material	Preparation	Extraction Technique	Solvent	Oil Yield (g/100 g DM Seed)	Ref.
Soybean Sunflower	Crushed seeds	Soxhlet extraction	α-Pinene	21.167.2	[97]
Rapeseed	Ground	Soxhlet extraction	*p*-Cymene	88.9 ^a^	[65]
			α-Pinene	65.5 ^a^	
			Limonene	80.8 ^a^	
Rapeseed	Coarsely ground	Soxhlet extraction	*p*-Cymene	39.7	[66]
			Limonene	37.0	
Rapeseed	Ground	Maceration with intense mixing	*cis*-Rich pinane (*cis/trans*: 7/3)	42.5	[98]
Rapeseed	Ground	Maceration with intense mixing Soxhlet extraction	*p*-Menthane	37.140.5	[99]

^a^ Oil extraction (% of total oil).

**Table 8 molecules-28-05973-t008:** Vegetable oils’ extraction with other organic solvents.

Material	Preparation	Extraction Technique	Solvent	Oil Yield (g/100 g DM Seed)	Ref.
Rapeseed	Coarsely ground	Soxhlet extraction	Ethyl acetate	42.8	[66]
			CPME	41.5	
			DMC	42.8	
Rapeseed Flaxseed	Ground	Pressurised solvent extraction	Ethyl acetate	40.4 33.3	[106]

**Table 9 molecules-28-05973-t009:** Vegetable oils’ extraction with aqueous enzymatic extraction.

Material	Preparation	Extraction Technique	Enzymes	Oil Extraction (% of Total Oil)	Protein Extraction (% of Total Protein)	Ref.
Soybean	Expanded	Thermo-controlled bath	Alcalase and Celluclast (adjusted pH)	26.8 ^a^	16.5	[113]
			Alcalase and Viscozyme L (adjusted pH)	29.5 ^a^	20.2	
			Alcalase and Celluclast	20.6 ^a^	7.3	
			Alcalase and Viscozyme L	20.2 ^a^	10.7	
Soybean	Dehulled, flaked and extruded	Thermo-controlled bath	Protex 51FP; Protex 6L or 7L	90–93	Nd	[114]
Soybean	Dehulled, flaked and extruded	Two-stage countercurrent extraction	Protex 6L	98	92	[112]
Rapeseed	Dehulled and cold-pressed	Thermo-controlled bath	Viscozyme L and Alcalase 2.4L	71.9	82.1	[115]
Canola	Ground	Thermo-controlled bath	Protex 7L	23.4 ^a^	5.9 ^a^	[116]
			Multifect Pectinase FE	22.2 ^a^	4.5 ^a^	
			Multifect CX 13L	26.0 ^a^	3.5 ^a^	
			Natuzyme	22.7 ^a^	4.3 ^a^	
Rapeseed	Dehulled	Thermo-controlled bath	Pectinase, Cellulase, and b-Glucanase	91.6, 73–76 ^b^	80–83	[117]
Sunflower	Ground	Thermo-controlled bath	Protex 7L	28.3 ^a^	4.3 ^a^	[118]
			Kemzyme	32.2 ^a^	2.4 ^a^	
			Alcalase 2.4L	26.6 ^a^	3.1 ^a^	
			Viscozyme L	39.7 ^a^	3.7 ^a^	
			Natuzyme	35.5 ^a^	2.7 ^a^	
Sunflower	Ground	Thermo-controlled bath	Celluclast 1.5L (Not buffered)	17.8 ^c^	Nd	[119]
			Celluclast 1.5L (buffered)	20.3 ^c^		
Sunflower	Ground	Thermo-controlled bath	Alcalase 2.4L	16.4 ^c^	Nd	[120]

^a^ Oil or protein yield (g/100 g DM seed); ^b^ free oil recovered (% of total oil); ^c^ free oil recovered (g/100 g DM seed).

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
