# Peer review of "Leading Edge Technologies and Perspectives in Industrial Oilseed Extraction†"

_molecules, 2023, doi:10.3390/molecules28165973_

Round 1

Reviewer 1 Report

In my opinion this is an interesting review. The information collected in sections 7 and 8 referring to “oilseed solvent extraction" is good. This is a quite comprehensive review. But the information in the previous sections requires a review, structuring and rewriting, more summarized and better organized.

Some General considerations are:

- Include an index of the sections included in the work

- Include a figure with the general scheme of the three phases for obtaining vegetable oils: oilseed storage and cleaning; oilseed preparation; oilseed extraction.

-Distribute the information as follows:

1) oilseed storage and cleaning

2) oilseed preparation

2.1…

2.2…

2.3…

3) oilseed extraction

3.1 oilseed mecanical extraction

3.1.1…

3.1.2..

3.2 oilseed sovent extraction

3.2.1. convectional solvents extraction

3.2.3. green solvent extraction

- Summarize the general procedures, put some schemes, production tables, etc.

-Keep the explanation of recent advances (change the name, because there are bibliographical references that are not so recent)

-Regarding the bibliographical references, remove the year, after the authors and keep only the number between brackets [1] [2]…

    some examples are: page 6, line185 “Murru and Lera Calvo (2020) is only [16], without specifying year. Page 9, line 311…

In Pag 26, in line 819: change : “extraction with hexane are presented below (table 3)” to: “extraction with hexane are presented in table 3.”

In page 34, in table 6. Why are there two different values in row 4, column 5?

In pag 38, “change ethyl acetate is a transparent ester..." to: “ethyl acetate is a colorless ester…”

-Review bibliography, in some citations the journal is missing. Example: page 44, nº 27

The English used is correct

Author Response

ANSWERS TO REVIEWER 1

Comments and Suggestions for Authors

In my opinion this is an interesting review. The information collected in sections 7 and 8 referring to “oilseed solvent extraction" is good. This is a quite comprehensive review. But the information in the previous sections requires a review, structuring and rewriting, more summarized and better organized.

All the authors warmly acknowledge the Reviewer for the positive assessment and the useful suggestions.

All the amendments have been corrected.

Some General considerations are:

- Include an index of the sections included in the work

Although we cannot see the index in the author rules if suggested it could be as follow:

Review index

  1. Introduction p. 1
  2. Oilseed storage and cleaning p. 2
  3. Oilseed preparation p. 4
  4. Oilseed extraction p. 13
  5. Green solvents extraction p. 26
  6. Conclusion and perspectives p. 43

- Include a figure with the general scheme of the three phases for obtaining vegetable oils: oilseed storage and cleaning; oilseed preparation; oilseed extraction.

The oilseed general processing is shown in Figure 2. This figure shows the common steps of receiving and preparation of the seeds, the overall process of seeds preparation and extraction, and the downstream processes to obtain the final product.

-Distribute the information as follows:

1) oilseed storage and cleaning

2) oilseed preparation

2.1…

2.2…

2.3…

3) oilseed extraction

3.1 oilseed mecanical extraction

3.1.1…

3.1.2..

3.2 oilseed sovent extraction

3.2.1. convectional solvents extraction

3.2.3. green solvent extraction

The text has been reorganised as requested.

- Summarize the general procedures, put some schemes, production tables, etc.

-Keep the explanation of recent advances (change the name, because there are bibliographical references that are not so recent)

The term “recent advances” has been removed following the reorganisation of the text as above.

-Regarding the bibliographical references, remove the year, after the authors and keep only the number between brackets [1] [2]…

    some examples are: page 6, line185 “Murru and Lera Calvo (2020) is only [16], without specifying year. Page 9, line 311…

All references have been modified accordingly.

In Pag 26, in line 819: change: “extraction with hexane are presented below (table 3)” to: “extraction with hexane are presented in table 3.”

The sentence was modified accordingly.

In page 34, in table 6. Why are there two different values in row 4, column 5?

In this table, extraction yields are given in three different measures: 1) g/100 g DM (dried matrix) seeds, 2) g/100 g initial material (not specified if dried), 3) % of total oil. For completeness, all results were given when more results were available. The yields given as "% of total oil" allow a direct comparison with hexane, which is normally used to determine the total oil content of seeds.

In pag 38, “change ethyl acetate is a transparent ester..." to: “ethyl acetate is a colorless ester…”

The sentence was modified accordingly.

-Review bibliography, in some citations the journal is missing. Example: page 44, nº 27

References have been checked.

Reviewer 2 Report

  The paper Leading Edge Technologies and Perspectives in Industrial Oilseeds Extraction by Christian Cravotto, Ombéline Claux, Mickaël Bartier, Anne-Sylvie Fabiano-Tixier and Silvia Tabasso presents the conventional methods of storage, processing and extraction of oilseeds, as well as innovative techniques of processing and extraction. In addition, the parameters that most affect yields and product quality at the industrial level are critically described. The extensive use of hexane for the extraction of most vegetable oils is undoubtedly the main concern of the entire production process in terms of health, safety and environmental issues. Special attention is paid to environmentally friendly solvents such as ethanol, supercritical CO2, 2-methyloxolanes, enzymatic water extraction, etc. The current state of the art in the use of green solvents is described and an objective assessment of their potential for more sustainable industrial processes is proposed. The authors have synthesized and processed with objectivity and critical spirit the information provided by the literature. The information provided by the authors is of real use to researchers in the field, The conclusions and perspectives are concise and to the point, based on the numerous specialized works consulted.

Author Response

ANSWERS TO REVIEWER 2

Comments and Suggestions for Authors

The paper Leading Edge Technologies and Perspectives in Industrial Oilseeds Extraction by Christian Cravotto, Ombéline Claux, Mickaël Bartier, Anne-Sylvie Fabiano-Tixier and Silvia Tabasso presents the conventional methods of storage, processing and extraction of oilseeds, as well as innovative techniques of processing and extraction. In addition, the parameters that most affect yields and product quality at the industrial level are critically described. The extensive use of hexane for the extraction of most vegetable oils is undoubtedly the main concern of the entire production process in terms of health, safety and environmental issues. Special attention is paid to environmentally friendly solvents such as ethanol, supercritical CO2, 2-methyloxolanes, enzymatic water extraction, etc. The current state of the art in the use of green solvents is described and an objective assessment of their potential for more sustainable industrial processes is proposed. The authors have synthesized and processed with objectivity and critical spirit the information provided by the literature. The information provided by the authors is of real use to researchers in the field, The conclusions and perspectives are concise and to the point, based on the numerous specialized works consulted.

All the authors warmly acknowledge the Reviewer for the positive assessment and the useful suggestions.

Reviewer 3 Report

This manuscript intensively reviewed the technology of oilseed extraction both of post-harvesting, pretreatment process and extraction (mechanical and solvent-based). Nor only focus on oil yield, this manuscript also explain how to preserve the quality of oil and protein content in defatted meal. It is very high quality. There are minor issues to improve the content of this manuscript.

1. Author should add more information regarding to purification process of crude oil.

2. For green solvent extraction, author already cover the aqueous extraction in enzymatic. Nevertheless, there are the other aqueous-based oil extraction such as aqueous two phase extraction by using polymer, aqueous-assisted surfactant oil extraction. Additional information regarding to these green solvent should include in this manuscript.

3. In aqueous extraction, author should provide more information regarding to the performance of microwave and ultrasonication technique on oil yield. As well as the demulsification process.

Author Response

ANSWERS TO REVIEWER 3

Comments and Suggestions for Authors

This manuscript intensively reviewed the technology of oilseed extraction both of post-harvesting, pretreatment process and extraction (mechanical and solvent-based). Nor only focus on oil yield, this manuscript also explain how to preserve the quality of oil and protein content in defatted meal. It is very high quality. There are minor issues to improve the content of this manuscript.

All the authors warmly acknowledge the Reviewer for the positive assessment and the useful suggestions.

  1. Author should add more information regarding to purification process of crude oil.

The purification process of mechanically extracted oils is described on page 15 line 476−488:

“The oil obtained by mechanical pressing usually contains a high concentration of meal fines (about 5 ̶ 10% by weight), which are removed in a screen kettle and then in a leaf or plate filter before the oil is sent to the refining process. In the traditional method, the oil is pumped into a tank where a residence time of 30 ̶ 60 minutes is observed to allow the heavier particles to settle and be removed from the bottom of the tank. After gravity separation, the oil is then pumped under pressure through a leaf filter to finally separate the fine particles.  Alternatively, a high-speed centrifugal decanter is used to separate the fine particles from the oil. The separated fine fraction, rich in oil, is generally recycled back into the process at the inlet of the press. A reduction in fines can be achieved by using expanders before pressing or in some cases after pressing to agglomerate the fines before the solvent extraction. After the centrifugal decanter, the prepress oil is typically in the range of 0.1% solids content and 0.2% moisture content. If the pre-press oil has to be degummed, it is generally mixed with the solvent-extracted oil and then degummed together. Whereas if it needs to be stored before further processing, it is generally passed through a vacuum dryer to reduce the moisture content below 0.1% and through a cooler to reduce the temperature below 50 °C before being pumped for storage [39].”

The general description of the crude oil refining process was added on page 22 line 703−718:

“Exept for virgin oils, crude oils cannot be consumed directly or used in various foods without technological refining processes. Crude oils such as soybean, rapeseed, palm, corn and sunflower oils must be purified or refined before consumption. The aim of refining is to obtain an odourless and rather neutral-tasting oil, limpid and colourless oil that is free of contaminants [51]. Compounds known to have a negative impact on the quality and stability of oils include free fatty acids, unsaponifiables, waxes, pigments, solid impurities (especially fibres), oxidation products (peroxides, aldehydes, ketones, and oxidised fatty acids). In addition, vegetable oils may also contain some contaminants: pesticides, trace metals, mineral oil aromatic hydrocarbons (MOAH), aflatoxins, dioxins, polycyclic aromatic hydrocarbons (PAH) and traces of organic solvents. However, one of the main disadvantages of refining is the loss of substances responsible for healthy and technological properties of the oils, such as tocopherols, phospholipids, squalene, polyphenols and phytosterols [52]. The two main industrial processes for vegetable oils refining are chemical and physical refining, as described in Figure 11.

Figure 11. Crude oil chemical and physical refining process: a general overview.

The difference between these two processes lies in the method used to remove the free fatty acids. In chemical refining, the free fatty acids are removed by adding caustic soda and separating the soap by centrifugation (mechanical separation), while in physical refining the free fatty acids and other compounds are removed in the final step by distillation under high vacuum with steam injection [51].”

For green solvent extraction, author already cover the aqueous extraction in enzymatic. Nevertheless, there are the other aqueous-based oil extraction such as aqueous two phase extraction by using polymer, aqueous-assisted surfactant oil extraction. Additional information regarding to these green solvent should include in this manuscript.

The section on aqueous oil extraction has been expanded to include the surfactant-assisted aqueous extraction. Page 42, lines 1359−1384

“Surfactant-aqueous oil extraction (SAOE) is another aqueous extraction technique that seems to be a promising method on a laboratory scale and allows an oil yield of slightly more than 90%. The basic concept of SAEP is to use surfactants to reduce the interfacial tension (IFT) between oil and water (about 19−24 mN/m), making aqueous oil extraction possible [122,123]. Surfactants reduce the IFT due to their amphiphilic structure, which gives them the property to adsorb at the oil-water interface. Compared to conventional surfactants, extended surfactants contain polar intermediate groups, such as polypropylene oxide and/or polyethylene oxide, inserted between the hydrophilic head and the lipophilic tail. This molecular structure gives surfactants the ability to achieve a very low IFT (≤ 10-2 mN/m). Lowering the IFT to such low values allows the formation of so-called microemulsions, which are ideal systems for the co-solubilisation of oil and water. In addition, salts (e.g. NaCl, CaCl2 and KCl) and alcohols (from ethanol to medium-chain alcohols such as octanol and decanol) are often added to surfactant solutions to improve their interfacial properties and thus reduce both the critical micelle concentration and the surface or interfacial tension. SAOE involves two main steps: extraction and separation. In the first step, the oil is extracted by immersing the matrix in the surfactant solution, usually with stirring. The oil and water form a turbid mixture that must be separated and purified to obtain a free oil phase. First, the solid phase is separated from the liquid phase by centrifugation. Then, in the liquid phase of interest, three layers are generally observed: (i) a layer of free oil, (ii) a microemulsion of oil in aqueous solution and (iii) the surfactant solution. In order to recover the oil from this liquid fraction, the system must be destabilised, whereby the oil-water -surfactant system is relatively stable. Gagnon et al. [122] described in detail the properties of extended surfactants, the state of the art of SAOE and the limitations that need to be overcome for the industrial application of this extraction method. The authors concluded that although initial results have shown this technology to be effective on a laboratory scale, it is still far from industrial application. From a technical point of view, the fractionation process of the oil-water-surfactant mixture, the recycling of the surfactants, the removal of salts and dehydration of the residual cake after extraction are the main critical points. In addition, the energy costs of the process should be taken into account and more information on the quality of the oil and solid fractions should be collected and compared with those obtained in the conventional way. Finally, the identification of new extended-like, bio-based and environmentally friendly surfactants would be desirable.”

  1. In aqueous extraction, author should provide more information regarding to the performance of microwave and ultrasonication technique on oil yield. As well as the demulsification process.

The authors described the use of microwave and ultrasound in the pre-treatment of seeds before extraction in section: 3.2. Microwave, ultrasound, and pulsed electric fields pre-treatments (p. 9).

A further study on the use of US before mechanical pressing of seeds has been added (p. 9, lines 299−305)

In the section on green solvents, the authors preferred to focus only on extraction solvents, without other process intensification technologies.

Reviewer 4 Report

It is a very well written article and represents a complete evaluation of the treatment of vegetable oils.

Being a study article of the specialized literature, it contains all existing procedures for treating and obtaining these types of raw materials.

I would recommend, if possible, a chapter dedicated to the results obtained following the economic and technical analysis of the analyzed extraction models. It can be added to this article or it can be part of another article that also contains the numerical equations and chemical reactions related to each process.

It is a good English writing,

Author Response

ANSWERS TO REVIEWER 4

Comments and Suggestions for Authors

It is a very well written article and represents a complete evaluation of the treatment of vegetable oils.

Being a study article of the specialized literature, it contains all existing procedures for treating and obtaining these types of raw materials.

I would recommend, if possible, a chapter dedicated to the results obtained following the economic and technical analysis of the analyzed extraction models. It can be added to this article or it can be part of another article that also contains the numerical equations and chemical reactions related to each process.

All the authors warmly acknowledge the Reviewer for the positive assessment and the useful suggestions.

For all the methods described, a technical and, where available, also an economic evaluation was carried out (e.g. p. 29 lines 925-931, 935−938 for Ethanol, p. 32 lines 1016−1023 for 2-MeOx, p. 35 lines 1139−1144 for SFE). A supplementary summary of both aspects would be interesting material for a future article.

Round 2

Reviewer 1 Report

Article can be accepted in its present form

Quality of English Language is good